# Coupled Multiwavelet Neural Operator Learning for Coupled Partial Differential Equations

**Xiongye Xiao**[1]*   **Defu Cao**[1]*   **Ruochen Yang**[1]   **Gaurav Gupta**[1]
**Gengshuo Liu**[1]   **Chenzhong Yin**[1]   **Radu Balan**[2]   **Paul Bogdan**[1]

[1] University of Southern California, Los Angeles, CA 90089, USA
[2] University of Maryland, College Park, MD 20742, USA

{xiongyex,defucao,ruocheny,ggaurav,gengshuo,chenzhoy,pbogdan}@usc.edu
rvbalan@umd.edu,

## Abstract

Coupled partial differential equations (PDEs) are key tasks in modeling the complex dynamics of many physical processes. Recently, neural operators have shown the ability to solve PDEs by learning the integral kernel directly in Fourier/Wavelet space, so the difficulty for solving the coupled PDEs depends on dealing with the coupled mappings between the functions. Towards this end, we propose a *coupled multiwavelets neural operator* (CMWNO) learning scheme by decoupling the coupled integral kernels during the multiwavelet decomposition and reconstruction procedures in the Wavelet space. The proposed model achieves significantly higher accuracy compared to previous learning-based solvers in solving the coupled PDEs including Gray-Scott (GS) equations and the non-local mean field game (MFG) problem. According to our experimental results, the proposed model exhibits a $2\times \sim 4\times$ improvement relative $L2$ error compared to the best results from the state-of-the-art models.

## 1 Introduction

Human perception relies on detecting and processing waves. While our eyes detect waves of electromagnetic radiation, our ears detect waves of compression in the surrounding air. Going beyond waves, from complex dynamics of blood flow to sustain tissue growth and life, to navigating underwater, ground and aerial vehicles at high speeds requires discovering, learning and controlling the partial differential equations (PDEs) governing individual or webs of biological, physical and chemical phenomena (Lacasse et al., 2007; Henriquez, 1993; Laval & Leclercq, 2013; Ghanavati et al., 2017; Radmanesh et al., 2020). Within this context, neural operators have been successfully used to learn and solve various PDEs. By representing the integral kernel termed as Green's function in the Fourier or Wavelet spaces, the fourier neural operator (Li et al., 2020b) and the multiwavelet-based neural operator (Gupta et al., 2021b;a)) exhibit significant improvements on solving PDEs compared with previous work. However, when it comes to coupled systems characterized by coupled differential equations such as mean field games (MFGs) (Lasry & Lions, 2007; Achdou & Capuzzo-Dolcetta, 2010), analysis of coupled cyber-physical systems (Xue & Bogdan (2017), or analysis of the surface currents in the tropical Pacific Ocean Bonjean & Lagerloef (2002), the interactions between the variables/functions need to be considered to decouple the system. Without the knowledge of underlying PDEs, the complex interactions can be hardly represented in the data-driven model. To build a data-driven model that can give a general representation of the interactions to efficiently solve coupled differential equations, we propose the coupled multiwavelets neural operator (CMWNO).

**Neural Operators.** Neural operators (Li et al., 2020b;c;a; Gupta et al., 2021b; Bhattacharya et al., 2020; Patel et al., 2021) focus on learning the mapping between infinite-dimensional spaces of functions. The critical feature for neural operators is to model the integral operator namely the Green's function through various neural network architectures. The graph neural operators (Li et al., 2020b;c) use the graph kernel to model the integral operator inspired by graph neural networks; the

---

*Equal Contribution

Fourier neural operator (Li et al., 2020b) uses an iterative architecture to learn the integral operator in Fourier space. The multiwavelet neural operators (Gupta et al., 2021b;a) utilize the non-standard form of the multiwavelets to represent the integral operator through 4 neural networks in the Wavelet space. The neural operators are completely data-driven and resolution independent by learning the mapping between the functions directly, which can achieve the state-of-the-art performance on solving PDEs and initial value problems (IVPs). To deal with coupled PDEs in the coupled system and be data-efficient, we aim to decode the various interaction scenarios inside the neural operators.

**Multiwavelet Transform.** In contrast to wavelets, multiwavalets (refer to Appendix C) use more than one scaling functions which are orthogonal. The multiwavelets exploit the advantages of wavelets, such as (*i*) the vanishing moments, (*ii*) the orthogonality, and (*iii*) the compact support. Along the essence of wavelet transform, a series of wavelet bases are introduced with scaled/shifted versions in multiwavelets to construct the basis of the coarsest scale polynomial subspace. The multiwavelet bases have been proved to be successful for representing integral operators as shown in (Alpert et al., 1993) (the discrete version of multiwavelets) and (Alpert, 1993b). In our proposed model, to develop compactly supported multiwavelets, we use the Legendre polynomials (Appendix D) which are non-zero only over a finite interval as the basis. The differential ($\partial/\partial t$) and the integral ($\iint_\Omega$) operators can be represented by the first-order multiwavelet coefficients ($s$ and $d$) of orthogonal bases via decomposition in the Wavelet space.

**Mean Field Games (MFGs).** As a representative problem for coupled systems in the real world, MFGs gains raising attentions in various areas, including economics (Achdou et al., 2014; 2022), finance (Guéant et al., 2011; Huang et al., 2019) and engineering (De Paola et al., 2019; Gomes et al., 2021), etc. Building on statistical mechanics principles and infusing them into the study of strategic decision making, MFGs investigate the dynamics of a large population of interacting agents seen as particles in a thermodynamic gas. Simply speaking, MFGs consist of (*i*) a Fokker–Planck equation (or related PDE) that describes the dynamics of the aggregate distribution of agents, which is coupled to (*ii*) a Hamilton–Jacobi–Bellman equation (another PDE) prescribing the optimal control of an individual (Lasry & Lions, 2006; 2007; Huang et al., 2006; 2007). Among different types of MFGs, the class of non-potential MFGs system with mixed couplings is particularly important as it is more reflective of the real world with a continuum of agents in a differential game.

**Solutions on MFGs.** Previous works either only restrict to systems without non-local coupling, such as alternating direction method of multipliers (ADMM) (Benamou & Carlier, 2015; Benamou et al., 2017) and primal-dual hybrid gradient (PDHG) algorithm (Briceno-Arias et al., 2019; 2018) or use general purpose numerical methods for solving the MFG problems (Achdou et al., 2013a;b; Achdou & Capuzzo-Dolcetta, 2010), which misses specific information from the target structure. In addition, the aforementioned works are not parallelizable with linear computational cost under the coupled MFGs settings. Recently, (Liu & Nurbekyan, 2020) considers dual variables of nonlocal couplings in Fourier or feature space. Furthermore, (Liu et al., 2021) expands the feature-space in the kernel-based representations of machine learning methods and uses expansion coefficients to decouple the mean field interactions. However, both dual variables and expansion coefficients need to bound the interactions of coupled system in a reasonable interval with prior knowledge. In our work, we first introduce the neural operator into coupled MFG fields, which can decouple the various interactions inside the multiwavelet domain.

**Novel Contributions.** The main novel contributions of our work are summarized as follows:

- For coupled differential equations, we propose a coupled neural operator learning scheme, named CMWNO. To the best of our knowledge, CMWNO is the first neural operator work using pure data-driven method to decouple and then solve coupled differential equations.

- Utilizing multiwavelet transform, CMWNO can deal with the interactions between the kernels of coupled differential equations in the Wavelet space. Specifically, we first yield the representation of coupled information during the decomposition process of multiwavelet transform. Then, the decoupled representation can interact separately to help the operators' reconstruction process. In addition, we propose a dice strategy to mimic the information interaction during the training process.

- The proposed model successfully learns the interaction between the coupled variables when the couple degree is increasing and thus it could open new directions for studying complex coupled systems via data-driven methods. Experimentally, the proposed CMWNO framework offers the state-of-the-art performance on both Gray-Scotts (GS) equations and non-local MFGs. Specifically, CMWNO outperforms the best baseline (MWT$_c$) by **54.0%** on GS equations with various resolutions and outperforms the best baseline (FNO$_c$) by **61.4%** on non-local MFGs with different time steps.

## 2 COUPLED MULTIWAVELET NEURAL OPERATORS LEARNING

To solve a coupled control system characterized by coupled state equations in control theory, a popular way is to use the Laplace operator $s$ to represent differential and integral operators (Gilbarg et al., 1977). Therefore, the coupled high-order differential equations can be transformed into the first-order differential equations in the Laplace space which will reduce the decoupling difficulty. Inspired by the use of the Laplace operator and the properties of the multiwavelets, we assume that the interactions between kernels can be used to approximate the coupled information by reducing the degree of high-order operators in multiwavelet bases. With this assumption, we are able to build the coupled multiwavelet neural operators (CMWNO) learning scheme, which utilizes decomposition representation from the operator and mimic the interaction via a dice strategy.

### 2.1 COUPLED DIFFERENTIAL EQUATIONS

To provide a simple example of the coupled kernels, $\kappa_1$ and $\kappa_2$, let us consider a general coupled system with 2 coupled variables $u(x, t)$ and $v(x, t)$ with the given initial conditions $u_0(x)$ and $v_0(x)$. Given $\mathcal{A}$ and $\mathcal{U}$ as two Sobolev spaces $\mathcal{H}^{s,p}$ with $s > 0, p = 2$, let $T$ denote a generic operator such that $T : \mathcal{A} \to \mathcal{U}$. Without the knowledge of how these two variables are coupled, to solve for $u(x, \tau)$ and $v(x, \tau)$, we need two operators $T_1$ and $T_2$ such that $T_1 u_0(x) = u(x, \tau)$ and $T_2 v_0(x) = v(x, \tau)$. The coupled kernels termed as Green's function can be written as follows:

$$
\begin{aligned}
T_1 u_0(x) &= \int_D \kappa_1(x, y, u_0(x), u_0(y), v_0(x), v_0(y), \kappa_2) u_0(y) dy, \\
T_2 v_0(x) &= \int_D \kappa_2(x, y, u_0(x), u_0(y), v_0(x), v_0(y), \kappa_1) v_0(y) dy, \\
u(x, 0) &= u_0(x); \quad v(x, 0) = v_0(x), \quad x \in D,
\end{aligned}
\tag{1}
$$

where $D \subset \mathbb{R}^d$ is a bounded domain. The interacted kernels cannot be directly solved without considering the interference from the other kernel, and our idea is to simplify the kernels first and deal with the interactions in the multiwavelet domain.

### 2.2 MULTIWAVELET OPERATOR

To briefly introduce the multiwavelet operator, we explain how the neural networks are used to represent the kernel in this section. The basic concept of multiresolution analysis (MRA) and multiwavelets (Alpert et al., 1993; Alpert, 1993a;b) are provided in the Appendix C.

**Notation** For $k \in \mathbb{Z}$ and $n \in \mathbb{N}$, the space of piecewise polynomial functions is defined as: $\mathbf{V}_n^k = \{f | \text{the restriction of } f \text{ to the interval } (2^{-n}l, 2^{-n}(l + 1)) \text{ is a polynomial of degree} < k,$ for all $l = 0, 1, \ldots, 2^n - 1$, and f vanishes elsewhere$\}$. $\mathbf{V}_0^k$ consists of the orthogonal scaling functions $\varphi_i$ with $i = 0, \ldots, k - 1$, and $\mathbf{V}_n^k$ can be spanned by shifting and scaling these functions as $\varphi_{jl}^n(x) = 2^{n/2} \varphi_j(2^n x - l)$, where $j = 0, \ldots, k - 1$ and $l = 0, \ldots, 2^n - 1$. The coefficients of $\varphi_{jl}^n(x)$ are called scaling coefficients marked as $s_{jl}^n$. The multiwavelet subspace $\mathbf{W}_n^k$ is defined as the orthogonal complement of $\mathbf{V}_n^k$ in $\mathbf{V}_{n+1}^k$ such that $\mathbf{V}_n^k \bigoplus \mathbf{W}_n^k = \mathbf{V}_{n+1}^k, \mathbf{V}_n^k \perp \mathbf{W}_n^k$. $\mathbf{W}_0^k$ consists of the orthogonal wavelet functions $\psi_i$ with $i = 0, \ldots, k - 1$. Similar to $\mathbf{V}_n^k$, $\mathbf{W}_n^k$ is composed of the wavelets functions $\psi_{jl}^n(x)$ with wavelets coefficients $d_{jl}^n$.

To represent the functions and learn the mapping in multiwavelet space, the nonstandard form is used to represent the integral operator. According to (Beylkin et al., 1991; Alpert et al., 2002b), an

orthogonal projection operator $P_n^k : \mathcal{H}^{s,2} \to \mathbf{V}_n^k$, and $Q_n^k : \mathcal{H}^{s,2} \to \mathbf{W}_n^k$ with $Q_n^k = P_{n+1}^k - P_n^k$, then an single operator $T$ in our coupled system can be represented as:

$$T = \bar{T}_0^k + \sum_{n=0}^{\infty}(A_n^k + B_n^k + C_n^k), \tag{2}$$

where $\bar{T}_0^k = P_0^k T P_0^k, A_n^k = Q_n^k T Q_n^k, B_n^k = Q_n^k T P_n^k, C_n^k = P_n^k T Q_n^k$, $Q_n^k$ is the multiwavelet operator. Therefore, the nonstandard forms of the operator is a collection of triplets $\{\bar{T}_0^k, (A_i^k, B_i^k, C_i^k)_{n=0,1,\ldots}\}$. For a given operator $T : Tu_0(x) = u_\tau(x)$, the map under wavelet space can be written as:

$$T_{d\,l}^i = A_i^k d_l^i + B_i^k s_l^i, \quad T_{\hat{s}\,l}^i = C_i^k d_l^i, \quad T_{s\,l}^0 = \bar{T} s_l^0, \quad i = 0, 1, \ldots, n \tag{3}$$

where, $(T_{s\,l}^i, T_{d\,l}^i)/(s_l^i, d_l^i)$ are the scaling/wavelet coefficients of $u_\tau(x)/u_0(x)$ in subspace $\mathbf{V}_{i+1}^k$. In our model, one kernel is approximated using 4 simple neural networks $A, B, C$ and $\bar{T}$ such that $T_{d\,l}^i \approx A_{\theta_A}(d_l^i) + B_{\theta_B}(s_l^i), T_{\hat{s}\,l}^i \approx C_{\theta_C}(d_l^i)$, and $T_{s\,l}^0 \approx \bar{T}_{\theta_{\bar{T}}}(s_l^L)$.

## 2.3 COUPLED MULTIWAVELETS MODEL

This section introduces a coupled multiwavelets model to provide a general solution on coupled differential equations. First, we make a mild assumption to decouple two coupled operators given in Section 2.1. To simplify eq. 1, without loss of generality, we assume that we can build two operators $T_u$ and $T_v$ to approximate $u(x, \tau)$ and $v(x, \tau)$, where $T_u$ and $T_v$ are decoupled and do not carry any interference from each other. In other words, we can write $T_u u_0(x) = u'(x, \tau); T_v v_0(x) = v'(x, \tau)$, where $u'(x, \tau)$ and $v'(x, \tau)$ are the approximations of $u(x, \tau)$ and $v(x, \tau)$ without coupling. The assumption is mild and easy to get satisfied in the Wavelet space since the operators can be represented by the first-order multiwavelet coefficients. According to this assumption, we can derive the following relations:

$$\begin{aligned} u(x, \tau) &= T_u u(x, 0) + \epsilon_1(T_v), \quad x \in D \\ v(x, \tau) &= T_v v(x, 0) + \epsilon_2(T_u), \quad x \in D \end{aligned} \tag{4}$$

where $\epsilon_1(T_u)$ quantifies the interference from operator $T_v$ to solve $u(x, \tau)$ and $\epsilon(T_v)$ represents the measurable interaction from operator $T_u$. Therefore, the integral operators can be written as:

$$T_u u_0(x) = \int_D \kappa_u(x, y) u_0(y) dy; \quad T_v v_0(x) = \int_D \kappa_v(x, y) v_0(y) dy, \tag{5}$$

the kernels $\kappa_u$ and $\kappa_v$ termed as Green's functions can be learned through neural operators, where $\kappa_u$ can be learned using the data of $u$ while the kernel $\kappa_v$ is learned from $v$. To model $\epsilon_1(T_u)$ and $\epsilon_2(T_v)$, we transform the operators into multiwavelet coefficients in the Wavelet space and embed it through simple linear combination after the decomposition steps.

Based on the concept of multiwavelets (Appendix Section C), here we simply explain the decomposition step and reconstruction step of multiwavelets in our coupled system. Since $\mathbf{V}_n^k = \mathbf{V}_{n-1}^k \bigoplus \mathbf{W}_{n-1}^k$ according to Section 2.2, the bases of $V_n^k$ can be written as a linear combination of the scaling functions $\varphi_i^{n-1}$ and the wavelet functions $\psi_i^{n-1}$. The linear coefficients $(H^{(0)}, H^{(1)}, G^{(0)}, G^{(1)})$ are termed as multiwavelet decomposition filters, transforming representation between subspaces $\mathbf{V}_{n-1}^k, \mathbf{W}_{n-1}^k$, and $\mathbf{V}_n^k$. For a given function $f(x)$, the scaling/wavelet coefficients $s_{jl}^n/d_{jl}^n$ of scaling/wavelet functions $\varphi_{jl}^n/\psi_{jl}^n$ are computed as:

$$s_{jl}^n = \int_{2^{-n}l}^{2^{-n}(l+1)} f(x)\varphi_{jl}^n(x)dx; \quad d_{jl}^n = \int_{2^{-n}l}^{2^{-n}(l+1)} f(x)\psi_{jl}^n(x)dx. \tag{6}$$

Using the multiwavelet decomposition filters, the relations between the coefficients on two consecutive levels $n$ and $n + 1$ are computed as (decomposition step):

$$\mathbf{s}_l^n = H^{(0)}\mathbf{s}_{2l}^{n+1} + H^{(1)}\mathbf{s}_{2l+1}^{n+1}; \quad \mathbf{d}_l^n = G^{(0)}\mathbf{s}_{2l}^{n+1} + G^{(1)}\mathbf{s}_{2l+1}^{n+1}. \tag{7}$$

Therefore, starting with the coefficients $s_l^n$, we repeatedly apply the decomposition step in eq. 7 to compute the scaling/wavelet coefficients on coarser levels. Similarly, the reconstruction step can be represented as:

$$\mathbf{s}_{2l}^{n+1} = H^{(0)\,T}\mathbf{s}_l^n + G^{(0)\,T}\mathbf{d}_l^n, \quad \mathbf{s}_{2l+1}^{n+1} = H^{(1)\,T}\mathbf{s}_l^n + G^{(1)\,T}\mathbf{d}_l^n. \tag{8}$$

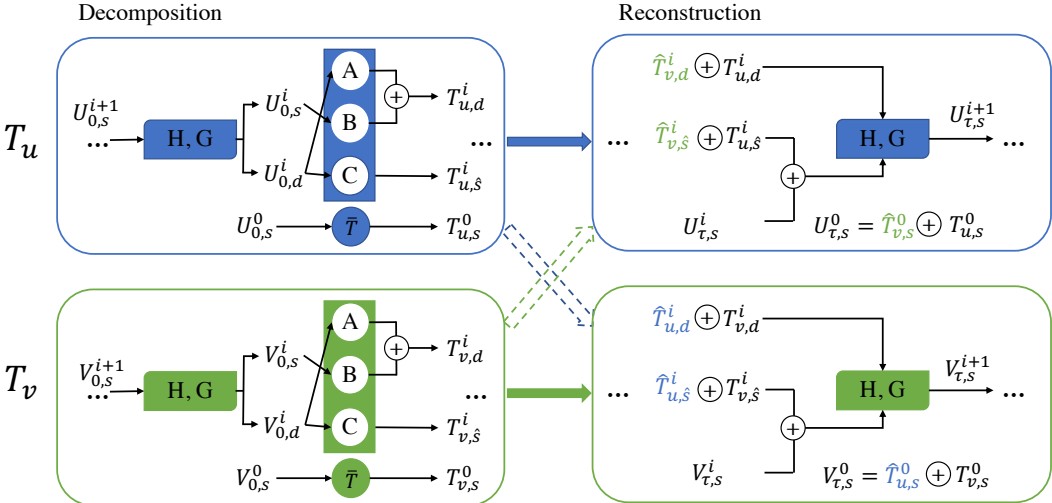

Figure 1: Architecture of CMWNO. Note that there are two coupled operators, $T_u$ and $T_v$, in our system, which aligns the number of coupled variables. The network $\bar{T}$ is only applied for the coarsest scale $L$ (0 in this system). The dashed arrows correspond to the auxiliary information from the unused operator without gradient during training process. For the interaction between operators, when we update the operator $T_u$, the decomposed ingredients from $T_v$ will be equipped into the reconstruction module of $T_u$ in the Wavelet domain, vice versa.

Repeatedly applying the reconstruction step, we can compute the coefficients $s_l^n$ from $s_l^0$ and $d_l^i, i = 0, \ldots, n$. In general, the function can be parameterized as the scaling/wavelet coefficients in the Wavelet space after the decomposition steps, and the coefficients can be mapped to the function after reconstruction steps. In our work, to model the interference $\epsilon_1(T_u)$ and $\epsilon_2(T_v)$, we obtain the multiwavelets coefficients of each kernel during the decomposition steps and embed them into the other kernel in the reconstruction step. Note that we will elaborate the detailed training strategy of how to mimic interactions inside our system in Section 2.4.

Our idea is to represent the functions and operators in Wavelet space to decouple the system using simple linear combinations. Considering the example in Section 2.1, according to the eq. 4 and 5, we first build two operators $T_u$ and $T_v$ such that $T_u u_0(x) = u_\tau'(x)$; $T_v v_0(x) = v_\tau'(x)$. For the operators $T_u$ and $T_v$, we denote their scaling/wavelet coefficients in wavelet domain as $T_{u,sl}^i$; $T_{u,dl}^i$ and $T_{v,sl}^i$; $T_{v,dl}^i$ respectively. For the input $u_0(x)$; $v_0(x)$ and the output $u_\tau(x)$; $v_\tau(x)$, we denote their coefficients as $U_{0,s(d)l}^i$; $V_{0,s(d)l}^i$ and $U_{\tau,s(d)l}^i$; $V_{\tau,s(d)l}^i$. According to eqs. 3 and 5, the multiwavelet coefficients of $T_u$ and $T_v$ can be calculated as:

$$T_{u,dl}^i = A_{u,i}^k U_{0,dl}^i + B_{u,i}^k U_{0,sl}^i, \quad T_{u,\hat{s}l}^i = C_{u,i}^k U_{0,dl}^i, \quad T_{u,sl}^0 = \bar{T} U_{0,sl}^0;$$
$$T_{v,dl}^i = A_{v,i}^k V_{0,dl}^i + B_{v,i}^k V_{0,sl}^i, \quad T_{v,\hat{s}l}^i = C_{v,i}^k V_{0,dl}^i, \quad T_{v,sl}^0 = \bar{T} V_{0,sl}^0, \tag{9}$$

where $i = 0, 1, \ldots, n$. Considering the interference from the other operators, the coefficients of the solutions $u_\tau(x)$ and $v_\tau(x)$ in the Wavelet space can be written as:

$$U_{\tau,dl}^i = T_{u,dl}^i + \hat{T}_{v,dl}^i, \quad U_{\tau,\hat{s}l}^i = T_{u,\hat{s}l}^i + \hat{T}_{v,\hat{s}l}^i, \quad U_{\tau,sl}^0 = T_{u,sl}^0 + \hat{T}_{v,sl}^0;$$
$$V_{\tau,dl}^i = T_{v,dl}^i + \hat{T}_{u,dl}^i, \quad V_{\tau,\hat{s}l}^i = T_{v,\hat{s}l}^i + \hat{T}_{u,\hat{s}l}^i, \quad V_{\tau,sl}^0 = T_{v,sl}^0 + \hat{T}_{u,sl}^0; \tag{10}$$

where $i = 0, 1, \ldots, n$. In the training process, the inputs of the neural networks $\{A_{[u,v]}, B_{[u,v]}, C_{[u,v]}, \bar{T}_{[u,v]}\}$ are the multiwavelet coefficients of $u_0(x)$; $v_0(x)$, and the outputs are the multiwavelet coefficients of $T_u$; $T_v$. When the neural networks $\{A_u, B_u, C_u, \bar{T}_u\}$ are trained for $T_u$, the neural networks $\{A_v, B_v, C_v, \bar{T}_v\}$ output $T_{v,[sl,dl]}^i$ without backpropagation, we use $\hat{T}_{[u,v],[sl,dl]}^i$ to mark the coefficients without gradient. Utilizing the orthogonality of the multi-wavelets, the coefficients embedding the information of the operators $T_u$; $T_v$ can be directly added to $T_v$; $T_u$ in the same Wavelet space $V_n^k$, then the neural networks with backpropagation can learn the

information from the other operator. In that way, the complex coupled equations can be solved via reducing the order of the functions and directly approximate decoupled functions at each iteration.

The architecture of the CMWNO is shown in Fig. 1, which illustrates the mapping process inside the wavelet space of layer $n$. The operations inside the wavelet space can be matched by the order of layers in the models, which means the decomposition operations for different resolutions are done independently. After decomposing $s^n$ via eq. 7, we can get the transferred information of input where each component will be used to reconstruct the original input at the layer $n$.

## 2.4 DICE STRATEGY

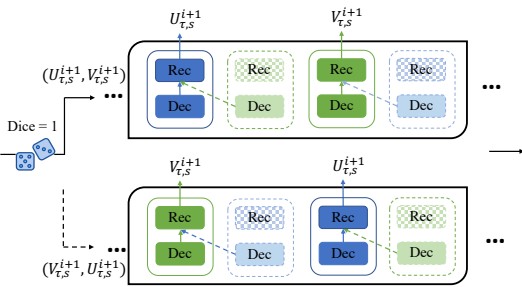

Figure 2: Dice strategy. For each sample, one only needs to go through a specific path (round diagonal corner rectangle). Inside each path, the order of updating is from left to right, where the darker block indicates the operator we want to update and the lighter blocks provide decomposition information from the fixed operator.

Inspired by scheduled sampling (Bengio et al., 2015) , which is designed to gently bridge the discrepancy between training and inference samples, we propose rolling the dice to randomly decide the interaction order between each neural operators, which is named dice strategy. Specifically, we roll the dice for every sample to decide which path to use, which can effectively mitigate the imbalance update problem for each kernel caused by the fixed training order. As illustrated in Fig. 2, when the dice tells the model to use path 1 (upper path) , we will update operator $T_u$ by equipping the coupled information from the other operator $T_v$ first. Note that, $T_v$ is learned by previous samples and have not updated yet. Then we use the updated operator $T_u$ to decompose the initial state $u_0$, which can be used to update $T_v$. Inside the Wavelet space with well-defined basis, where we are able to utilize vary orthogonal information from each initial state jointly. Note that this strategy is scalable to more operators referring to Fig. 6 in Supplementary and we left the design of this strategy for future work.

## 3 EXPERIMENTS

In this section, we empirically evaluate the proposed model on famous coupled PDEs such as the Gray-Scott (GS) equations and the non-local mean field game (MFG) problem characterized by coupled PDEs. Note that we compare against the state-of-the-art data-driven models which fits for our research goal to build efficient coupled operators for general downstream data-driven applications without sufficient expert knowledge. The experiments show that CMWNO not only achieves the lowest $L2$ relative errors when solving coupled PDEs, but also works consistently great under different input conditions. For the data structure, since our datasets are functions, we apply point-wise evaluations on the input and output data. For example, for the function $f(x), x \in D$, we discretize the domain as $x_1, \ldots, x_s \in D$, where $x_i$ are s-point discretization of the domain. Unless stated otherwise, we train on 1000 samples and test on 200 samples.

**Model architecture.** In our proposed model, for each operator, the neural networks $A$, $B$ and $C$ use a single-layered convolutional neural networks while $T$ uses a single linear layer. Our model is extensible and each kernel constructed by 4 neural networks $\{A, B, C, \bar{T}\}$ learning the mapping in wavelet space. The number of the kernels can be chosen based on the number of coupled variables or the number of explicit operators.

**Benchmark models.** We compare our model with the state-of-the-art neural operators including Fourier neural operator (**FNO**), Multiwavelet-based neural operator (**MWT**), and Padé exponential model (**Padé**), which show the best performance on solving PDEs according to the experiment results in (Li et al., 2020b;a; Gupta et al., 2021b;a). For the benchmark neural operator models, since we have the coupled functions as input and output (e.g., $u$ and $v$), we concatenate $u$ and $v$ for the models and marked the models as $FNO_c$, $MWT_c$, and $Padé_c$. We also use two single multiwavelet-based

neural operators to learn $u_\tau(x)$; $v_\tau(x)$ from $v_\tau(x)$; $u_\tau(x)$ independently and mark the model as $MWT_s$.

Similar to the coupling structure of our CMWNO, by creating the multiple kernels learned in Fourier space and applying the dice strategy during the Fourier transform, we build the coupled Fourier neural operator and mark it as CFNO.

**Training parameters.** The neural operators are trained using Adam optimizer with a learning rate of 0.001 and decay of 0.95 after every 100 steps. The models are trained for a total of 500 epochs which is the same with training CMWNO for fair comparison. All experiments are done on an Nvidia $A$100 40GB GPUs.

## 3.1 GRAY-SCOTT (GS) EQUATIONS

The GS equations are coupled differential equations which model the underlying reaction and diffusion patterns of chemical species. It is also able to generate a wide range of patterns which exist in nature, such as bacteria, spirals and coral patterns. Each variable (i.e., $u$ and $v$) diffuses independently with a linear growth or decay term, while coupled together by $\pm uv^2$ (Trefethen & Embree, 2001; Driscoll et al., 2014). For a given field $u(x,t)$; $v(x,t)$, the GS equations take the form:

$$\partial_t u(x,t) = \epsilon_1 \partial_{xx} u(x,t) + F(1 - u(x,t)) - \lambda u(x,t)v^2(x,t), \quad x \in (0,10), t \in (0,1]$$
$$\partial_t v(x,t) = \epsilon_2 \partial_{xx} v(x,t) - (K + F)v(x,t) + \lambda u(x,t)v^2(x,t), \quad x \in (0,10), t \in (0,1] \quad (11)$$
$$u(x,0) = u_0(x); \quad v(x,0) = v_0(x), \quad x \in (0,10)$$

where $\epsilon_1 = 1, \epsilon_2 = 10^{-2}, K = 6.62 \times 10^{-2}, F = 2 \times 10^{-2}$. We use the coupling coefficient $\lambda \in (0,1]$ to control the degree of coupling of $u$ and $v$. We aim to learn the operators (i) mapping the initial condition $u(x,0)$ to the solution $u(x, t = 1)$ with the interference of $v(x,t)$; (ii) mapping the initial condition $v(x,0)$ to the solution $v(x, t = 1)$ considering the interference of $u(x,t)$. The initial conditions are generated in Gaussian random fields (GRF) according to $u_0(x), v_0(x) \sim \mathcal{N}(0, 7^4(-\Delta + 7^2 I)^{-2.5})$ with periodic boundary conditions. We also use a different scheme to generate $u_0(x)$ by using the smooth random functions (Rand) in *chebfun* package (Driscoll et al., 2014) which returns a band-limited function defined by a Fourier series with independent random coefficients; the parameter $\gamma$ specifies the minimal wavelength and here we choose $\gamma = 0.5$. Therefore, generating the initial conditions by different schemes, we have two combinations of the initial conditions (i.e., $u_0(x)$ and $v_0(x)$) and we mark them as (U-GRF, V-GRF) and (U-Rand, V-GRF) respectively according to the generating schemes. Given the initial conditions, we solve the equations using a fourth-order stiff time-stepping scheme named as *ETDRK4* (Cox & Matthews, 2002) with a resolution of $2^{10}$, and sub-sample this data to obtain the datasets with the lower resolutions.

| Models | s=256 | | s=512 | | s=1024 | |
|--------|-------|-------|-------|-------|--------|-------|
| | u | v | u | v | u | v |
| CMWNO | **0.00468** | **0.00464** | **0.00492** | **0.00434** | **0.00471** | **0.00450** |
| CFNO | 0.01371 | 0.00654 | 0.01345 | 0.00643 | 0.01421 | 0.00619 |
| $MWT_s$ | 0.08075 | 0.07308 | 0.08041 | 0.07382 | 0.07996 | 0.07213 |
| $MWT_c$ | 0.01445 | 0.00742 | 0.01408 | 0.00744 | 0.01334 | 0.00779 |
| $FNO_c$ | 0.01431 | 0.00812 | 0.01542 | 0.00819 | 0.01545 | 0.00885 |
| $Padé_c$ | 0.01904 | 0.00964 | 0.02070 | 0.01022 | 0.02233 | 0.01055 |

Table 1: Gray–Scott (GS) equation benchmarks for different input resolution $s$ at initial condition (U-GRF,V-GRF). The relative $L2$ errors are shown for each model. **Bolded** values are the best results of all the models, and underlined values are the best results of the existing models. Set the same below.

**Varying resolution.** The results of our experiments on GS equations with different resolutions (i.e., $s = 256, 512, 1024$) are shown in Table 1. As shown in the results, all the models exhibit the resolution independence. The model $MWT_c$ with concatenated data performs better than model $MWT_s$ using two independent single MWT models to train $u$ and $v$ separately, which indicates the information from $v_0(x)$; $u_0(x)$ benefits the model predicting for $u_\tau(x)$; $v_\tau(x)$. For solving $u_\tau(x)$; $v_\tau(x)$, our proposed CMWNO outperforms $2X - 3X$ improvements compared with the best benchmark with respect to relative $L2$ error. CFNO also outperforms $FNO_c$, however, compared

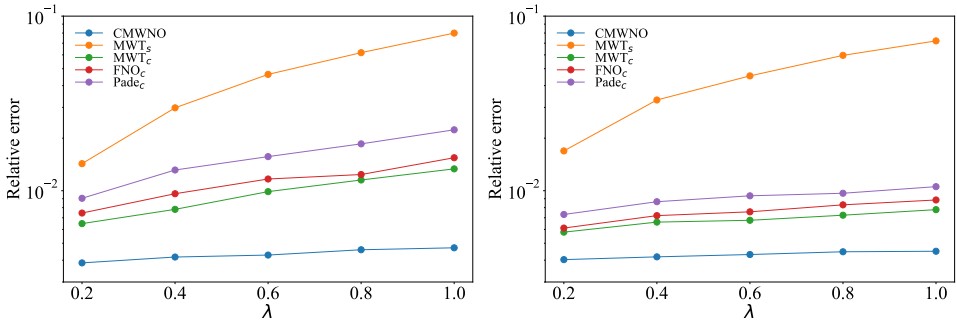

Figure 4: Comparing the models by varying the coupling coefficient $\lambda$ at the initial condition (U-GRF, V-GRF) with resolution $s = 1024$.

with the improvement of CMWNO on MWT, the improvement of CFNO on FNO is not significant, indicating that decoupling in the Fourier space is not as efficient as decoupling in the multiwavelets domain. The learning curve of the neural operators solving $u_\tau(x)$ at resolution $s = 1024$ is shown in Fig. 3.

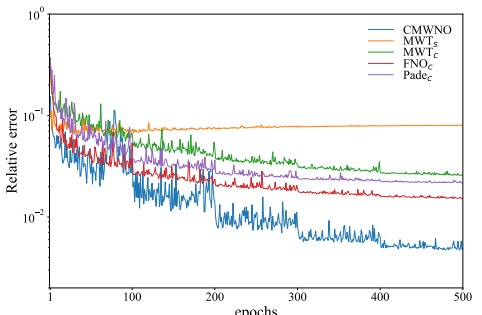

Figure 3: Learning curve - Relative $L2$ error $vs$ epochs for neural operators.

**Varying coupling coefficient** By varying the coupling coefficient $\lambda$ in the GS equations, we can get different degree of coupling between $u$ and $v$ according to eq. 11. The higher value of $\lambda$ means higher degree of coupling between $u$ and $v$. Given the same initial conditions $u_0(x)$ and $v_0(x)$, the outputs with different $\lambda$ (i.e., $\lambda = 0.2, 0.4, 0.6, 0.8, 1$) are shown in Fig. 4. It shows that as $\lambda$ increases, all the models perform worse. For solving $u_\tau(x)$, compared with at $\lambda = 0.2$, the relative $L2$ errors at $\lambda = 1$ of the models increase by **22.0% (CMWNO)**; 459.6% (MWT$_s$);

105.9% (MWT$_c$); 107.4% (FNO$_c$); 146.7% (Padé$_s$). In terms of $v_{(x,\tau)}$, the numbers are **11.6% (CMWNO)**; 326.8% (MWT$_s$); 34.5% (MWT$_c$); 44.8% (FNO$_c$); 44.3% (Padé$_s$). As we can see, the MWT$_s$ works the worst since the model cannot learn the interaction between $u$ and $v$. The models learning coupled operators through concatenated data works better than the single model but still do not perform well on high coupling data. On the contrary, our CMWNO outperforms well consistently with both low / high coupling coefficient, which indicates that our architecture is able to decouple the coupled kernels.

**Varying initial conditions** In addition to experimenting with both the initial conditions $u_0(x)$ and $v_0(x)$ generated in the GRF as marked (U-GRF, V-GRF), we also perform the models on (U-Rand,V-GRF). The numerical results are shown in Table 3 (see Appendix F). Our CMWNO achieves the lowest relative $L2$ error on both $u$ and $v$ with $3X$ and $2X$ improvements respectively. We provide a sample of initial conditions in Fig.7 (see Appendix E), and Fig. 5 shows its predicted outputs from models CMWNO, MWT$_s$ and MWT$_c$. It shows that our proposed CMWNO can give a precise prediction in a smooth way while MWT$_s$ and MWT$_c$ can only fit the true curve roughly.

| Models | t=0.2 | | t=0.4 | | t=0.6 | | t=0.8 | |
|---|---|---|---|---|---|---|---|---|
| | $\rho$ | $\phi$ | $\rho$ | $\phi$ | $\rho$ | $\phi$ | $\rho$ | $\phi$ |
| CMWNO | **0.00083** | **0.00073** | **0.00154** | **0.00252** | **0.00543** | **0.00467** | **0.02417** | **0.00305** |
| CFNO | 0.00767 | 0.00162 | 0.00890 | 0.00638 | 0.02011 | 0.01029 | 0.04780 | 0.00744 |
| MWT$_c$ | 0.00328 | 0.00646 | 0.00916 | 0.02244 | 0.02245 | 0.02768 | 0.06011 | 0.01622 |
| FNO$_c$ | 0.00241 | 0.00278 | 0.00473 | 0.00667 | 0.01329 | 0.01096 | 0.04950 | 0.00818 |
| Padé$_c$ | 0.00213 | 0.00320 | 0.00473 | 0.01307 | 0.01189 | 0.02466 | 0.03676 | 0.01171 |

Table 2: The relative $L2$ errors for predicting $\rho(x,t)/\varphi(x,t)$ with $t = 0.2, 0.4, 0.6,$ and $0.8$.

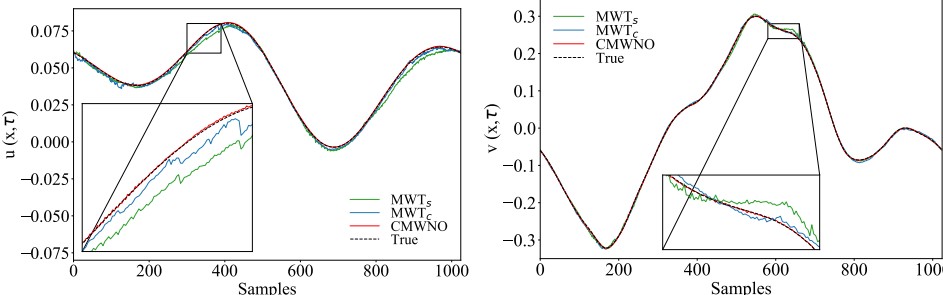

Figure 5: The output of GS couple equations at the initial condition (U-Rand, V-GRF). (Left) The predicted output of the models to $u(x, \tau = 1)$. (Right) The predicted output of the models to $v(x, \tau = 1)$.

## 3.2 MEAN FIELD GAME PROBLEM

For local interactions, directly discretizing interaction terms is economical. However, non-local MFG requires each player in making decisions to take into account the global information rather than local information, which will increase the amount of computation in the process of calculation. In other words, we need matrix multiplication on a full grid to calculate the interaction terms by evaluating the expressions $\int_\omega K(x, y)\rho(y, t)dy$. In this work, we propose a more general framework, CMWNO, to model the interactions in the Wavelet space and the results show that our model can be used to deal with the coupled systems. Here we solve the non-local MFG which can be characterized as:

$$\partial_t \rho(x, t) + \nabla \cdot (\rho(x, t)\nabla\varphi(x, t)) = 0, \quad x \in [0, 1], t \in (0, 1)$$
$$\partial_t \varphi(x, t) - \frac{1}{2}\|\varphi(x, t)\|^2 + \int_D K(x, y)\rho(y, t)dt = 0, \quad x \in [0, 1], t \in (0, 1) \tag{12}$$

where $\rho(x, t)$ is the density distribution of the players, and $\varphi(x, t)$ is the cost function. In a forward-forward MFG setting (Gomes & Sedjro, 2017), we can obtain the value of $\rho(x, 0)$ and $\varphi(x, 0)$. We aim to learn the operators: (i) mapping the initial condition $\rho(x, 0)$ to the solution $\rho(x, t = \tau)$ with the interference of $\varphi(x, t)$; (ii) mapping the initial condition $\varphi(x, 0)$ to the solution $\varphi(x, t = \tau)$ considering the interference of $\rho(x, t)$. To obtain the datasets, we generate $\rho(x, 0); \rho(x, t = 1)$ by using the random functions in *chebfun* package with the wavelength parameter $\gamma = 0.3; 0.1$, respectively. The coupled equations are numerically solved by the primal-dual hybrid gradient (PDHG) algorithm (Briceno-Arias et al., 2019; 2018) with the resolution $s = 256$. The initial conditions of $\rho(x, 0)$ and $\varphi(x, 0)$ are used as the input while the $\rho(x, t)$ and $\varphi(x, t)$ ($t = 0.2, 0.4, 0.6, 0.8$) are taken as the output.

We perform all the models working for coupled datasets mentioned above to solve this MFG coupled PDEs, and the results with different $t$ are shown in Table 2. Compared to the existing model with the best results, our proposed CMWNO yields $34.2\% \sim 67.4\%$ improvements in terms of $\rho$ and $57.4\% \sim 73.7\%$ in terms of $\varphi$ with respect to the relative $L2$ error. It is worth noting that $\text{MWT}_c$ performs the worst in most cases which indicates that the interactions between $\rho$ and $\varphi$ can not be learned through a single multiwavelet kernel. By interacting two kernels in the Wavelet space after decomposition steps, our proposed CMWNO can better decouple the interactions between $\rho$ and $\varphi$ to solve the MFG PDEs.

## 4 CONCLUSION

In this work, we propose a coupled multiwavelets neural operator using multiwavelet discretization of the spatial domain. Solving for coupled equations requires an information entanglement across operators for individual process. We found that combining operators in the projected domain of multiwavelets is effective. Numerical experiments using representative coupled PDEs including Gray-Scott and mean field game problem show that our coupling mechanism effectively learns the two processes in comparison with standalone operators.

ACKNOWLEDGEMENT

We are thankful to the anonymous reviewers for providing their valuable feedback which improved our manuscript. We would also like to thank Dr. Justinian Rosca and Lisang Ding for their valuable feedback. We gratefully acknowledge the support by the National Science Foundation Career award under Grant No. Cyber-Physical Systems / CNS-1453860, the NSF award under Grant CCF-1837131, MCB-1936775, CNS-1932620, the U.S. Army Research Office (ARO), the Defense Advanced Research Projects Agency (DARPA) Young Faculty Award and DARPA Director Award under Grant No. N66001-17-1-4044, an Intel faculty award, the Okawa Foundation award, a Northrop Grumman grant, and Google cloud program. A part of this work used the Extreme Science and Engineering Discovery Environment (XSEDE), which is supported by National Science Foundation grant number ACI-1548562. R.B. has been supported in part by a NSF award under grant DMS-2108900 and by the Simons Foundation. The views, opinions, and/or findings contained in this article are those of the authors and should not be interpreted as representing the official views or policies, either expressed or implied by the Defense Advanced Research Projects Agency, the Army Research Office, the Department of Defense, or the National Science Foundation.

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

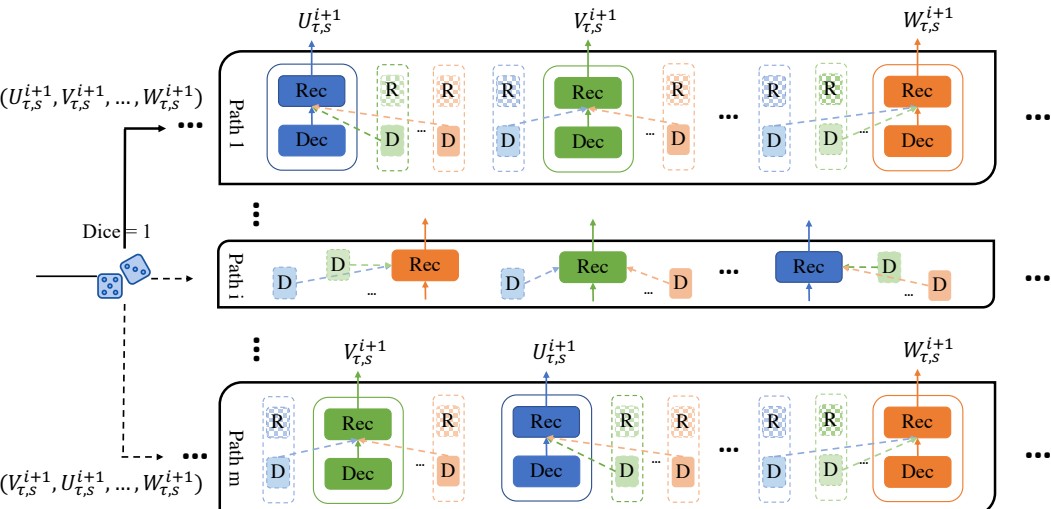

Figure 6: **The scalable dice strategy with multiple coupled kernels.** For each sample of coupled variables, one only needs to go through a specific path (round diagonal corner rectangle). For example, we get dice equal to 1 in this case so that this specific sample will go through Path 1. Inside each path, the order of updating is from left to right, where the darker block indicates the operator we want to update and the lighter blocks provide decomposition information from the fixed operator. Note that the updated operator can help to update other operators at the same stage that occurs after the specific operator. Thus, for $n$ kernels, we have $n$ operators that need to be updated and $m$ paths that can be selected, where $m = \mathcal{A}_n^n$ and $\mathcal{A}$ is the function of all permutations.

## A    RELATED WORK

The operator network can be made up of several neural networks, which aims to approximate any function defined as an input in one network and evaluated at the locations specified in the target network (Lu et al., 2022). Chen & Chen propose the universal approximation theory of operators for a single layer, which has led to several research works recently. Among them, DeepONet (Lu et al., 2021) first applies deep neural network on the universal approximation theorem to learn nonlinear operators. After that, Fourier Neural Operator (FNO) formulates the operator regression by parameterizing the integral kernel directly in Fourier space (Li et al., 2020a). Several works take advantage of FNO's ability to efficiently solve PDEs to design models for applications in practical chaotic systems such as turbulence simulation (Stachenfeld et al., 2021), multiphase flow simulation (Wen et al., 2022), and weather forecasting (Pathak et al., 2022). In addition, we would like to highlight that neural operators are able to tackle the more fundamental problems in time series analysis (Cao et al., 2021; Zhang et al., 2022; Cao et al., 2020) and easily to finds many application in transportation, healthcare, manufacturing, finance (Cao et al., 2022), etc. Gupta et al. introduce a multiwavelet-based neural operator that compresses the associated operator's kernel using fine-grained wavelets and the same group further proposes non-linear operator approximation for initial value problems. Besides, physics-informed neural network and machine learning methods provide a new research direction to equip specific physics information into neural operators Goswami et al. (2022); Meng et al. (2022). For more information of neural operators, please refer to the new survey paper: (Kovachki et al., 2021). Another related research line of our work is coupled PDEs Tang et al. (2009), which is usually discussed in the form of MFGs (Benamou & Carlier, 2015; Benamou et al., 2017; Briceno-Arias et al., 2019; 2018; Liu & Nurbekyan, 2020; Liu et al., 2021). In addition, the interaction between terms in complex systems Xue & Bogdan (2017); Xiao et al. (2021); Yin et al. (2020) can be characterized by coupled PDEs. Combining those two research lines, this is the first work which proposes a decoupled multiwavelet-based neural operator learning schema to solve coupled PDE problems.

## B    REPRODUCIBILITY & CODE AVAILABILITY

**Architecture description in detail**: The CMWNO model in Figure 1 is presented in the form of a recurrent cell. In the decomposition stage, the input data at each iteration are $U_{0,s}^{(i+1)}$ $(V_{0,s}^{(i+1)})$ which then gets transformed into $U_{0,s}^i, U_{0,d}^i$ $(V_{0,s}^i, V_{0,d}^i)$ using the filters $H, G$. At the same iteration, we also obtain the corresponding outputs $T_{u,d}^i$ and $T_{u,\hat{s}}^i$ $(T_{v,d}^i$ and $T_{v,\hat{s}}^i)$. The same process is repeated in the next iteration but now with using $U_{0,s}^i$ $(V_{0,s}^i)$ (obtained from the previous step) as the input, thus this makes a recurrent chain of operations and is a kind of ladder-down operation. The loop is repeated till we reach the $L$-th scale (coarsest scale) at which the final operation of $\bar{T}$ is applied according to eqs 9. The trainable neural networks layer in this stage is composed of $\{A_u, B_u, C_u, \bar{T}_u\}$ and $\{A_v, B_v, C_v, \bar{T}_v\}$. We use two cascading neural network layers with normal Xavier initialization to handle 1D coupled equations in our experiments. Moreover, $\{A_*, B_*, C_*\}$ are all one-layer CNNs with the ReLU Nair & Hinton (2010) activation function followed by a linear layer, where CNN's kernel size equals 3, stride equals 1, and padding size equals 1. The input channel of CNN equals the feature number and the output channel is set to be 128 in all the experiments. In addition, $\{\bar{T}_v\}$ is a single $k \times k$ linear layer with $k = 4$ suggested by Gupta et al. (2021b). In the reconstruction stage (which is ladder-up), iteratively, the outputs of decomposition part $T_{u,d}^i$ and $T_{u,\hat{s}}^i$ $(T_{v,d}^i$ and $T_{v,\hat{s}}^i)$ are first combined by the dice strategy in section 2.4, and then we use a reconstruction filter $H, G$. to obtain the finer scales $U_{\tau,s}^{(i+1)}$ $(V_{\tau,s}^{(i+1)})$, and finally, $U_{\tau,s}^n$ $(V_{\tau,s}^n)$ is the finest scale of the output.

Our code to run the experiments can be found at `https://github.com/joshuaxiao98/CMWNO/`.

## C    MULTIWAVELET BASES

### C.1    MULTIRESOLUTION ANALYSIS

The basic idea of MRA is to establish a preliminary basis in a subspace $V_0$ of $L^2(\mathbb{R})$, and then use simple scaling and translation transformations to expand the basis of the subspace $V_0$ into $L^2(\mathbb{R})$ for analysis on multiscales. Multiwavelets further this operation by using a class of orthogonal polynomials (OPs), in our case, we use Legendre polynomials for an efficient representation over a finite interval (Alpert et al., 2002a).

For $k \in \mathbb{Z}$ and $n \in \mathbb{N}$, the space of piecewise polynomial functions is defined as: $\mathbf{V}_n^k = \{f|$the restriction of $f$ to the interval $(2^{-n}l, 2^{-n}(l + 1))$ is a polynomial of degree $< k$, for all $l = 0, 1, \ldots, 2^n - 1$, and f vanishes elsewhere$\}$. Therefore, the space $\mathbf{V}_n^k$ has dimension $2^n k$, and each subspace $\mathbf{V}_i^k$ is contained in $\mathbf{V}_{i+1}^k$ shown as $\mathbf{V}_0^k \subset \mathbf{V}_1^k \subset \ldots \mathbf{V}_n^k \subset \ldots$. Given a basis $\varphi_0, \varphi_1, \ldots, \varphi_{k-1}$ of $\mathbf{V}_0^k$, the space $\mathbf{V}_n^k$ is spanned by $2^n k$ functions obtained from $\varphi_0, \varphi_1, \ldots, \varphi_{k-1}$ of $\mathbf{V}_0^k$ by shifts and scales as

$$\varphi_{jl}^n(x) = 2^{n/2}\varphi_j(2^n x - l), \quad j = 0, 1, \ldots, k - 1, \quad l = 0, 1, \ldots, 2^n - 1. \tag{13}$$

The functions $\varphi_0, \varphi_1, \ldots, \varphi_{k-1}$ are also called scaling functions which can project a function to the approach space $\mathbf{V}_0^k$.

### C.2    MULTIWAVELETS

The multiwavelet subspace $\mathbf{W}_n^k$ is defined as the orthogonal complement of $\mathbf{V}_n^k$ in $\mathbf{V}_{n+1}^k$, such that

$$\mathbf{V}_n^k \bigoplus \mathbf{W}_n^k = \mathbf{V}_{n+1}^k, \quad \mathbf{V}_n^k \perp \mathbf{W}_n^k; \tag{14}$$

and $\mathbf{W}_n^k$ has dimension $2^n k$. Therefore, the decomposition can be obtained as

$$\mathbf{V}_n^k = \mathbf{V}_0^k \bigoplus \mathbf{W}_0^k \bigoplus \mathbf{W}_1^k \ldots \bigoplus \mathbf{W}_{n-1}^k. \tag{15}$$

To form the orthogonal bases for $\mathbf{W}_n^k$, a class of bases is constructed for $L^2(\mathbb{R})$. Each basis consists of translates and dilates of a finite set of functions $\psi_1, \ldots \psi_k$ shown as follows:

$$\psi_{jl}^n(x) = 2^{n/2}\psi_j(2^n x - l), \quad j = 0, 1, \ldots, k - 1, \quad l = 0, 1, \ldots, 2^n - 1. \tag{16}$$

where the wavelet functions $\psi_1, \ldots \psi_k$ are piecewise polynomial and orthogonal to low-order polynomials (vanishing moments):

$$\int_0^1 x^i \psi_j(x) dx = 0, \quad i = 0, 1, \ldots, k-1. \tag{17}$$

Here we restrict our attention to the interval $[0, 1] \in \mathbb{R}$; however, the transformation to any finite interval $[p, q]$ could be directly obtained by the appropriate translates and dilates.

## D  LEGENDRE POLYNOMIALS

The Legendre polynomials are defined with respect to (w.r.t.) a uniform weight function $w_L(x) = 1$ for $-1 \leqslant x \leqslant 1$ or $w_L(x) = \mathbf{1}_{[-1,1]}(x)$ such that

$$\int_{-1}^1 P_i(x) P_j(x) dx = \begin{cases} \frac{2}{2i+1} & i = j, \\ 0 & i \neq j. \end{cases} \tag{18}$$

For our work, we shift and scale the Legendre polynomials so they are defined over $[0, 1]$ as $P_i(2x - 1)$, and the corresponding weight function as $w_L(2x - 1)$. The Legendre polynomials satisfy the following recurrence relationships

$$i P_i(x) = (2i - 1)x P_{i-1}(x) - (i - 1)P_{i-2}(x), (2i + 1)P_i(x) \qquad = P'_{i+1}(x) - P'_{i-1}(x),$$

which allows the expression of derivatives as a linear combination of lower-degree polynomials itself as follows:

$$P'_i(x) = (2i - 1)P_{i-1}(x) + (2i - 3)P_{i-1}(x) + \ldots, \tag{19}$$

where the summation ends at either $P_0(x)$ or $P_1(x)$, with $P_0(x) = 1$ and $P_1(x) = x$.

A set of orthonormal basis of the space of polynomials with degree $< d$ defined over the interval $[0, 1]$ is obtained using shifted Legendre polynomials such that

$$\phi_i = \sqrt{2i + 1} P_i(2x - 1),$$

w.r.t. weight function $w(x) = w_L(2x - 1)$, such that

$$\langle \phi_i, \phi_j \rangle_\mu = \int_0^1 \phi_i(x)\phi_j(x) dx = \delta_{ij}.$$

The basis for $V_0^k$ are chosen as normalized shifted Legendre polynomials of degree upto $k$ w.r.t. weight function $w_L(2x - 1) = \mathbf{1}_{[0,1]}(x)$ from Section D. For example, the first three bases are

$$\begin{aligned} \phi_0(x) &= 1, \\ \phi_1(x) &= \sqrt{3}(2x - 1), \\ \phi_2(x) &= \sqrt{5}(6x^2 - 6x + 1), \quad 0 \leqslant x \leqslant 1. \end{aligned} \tag{20}$$

For deriving a set of basis $\psi_i$ of $W_0^k$ using GSO, we need to evaluate the integrals efficiently, which could be achieved using the Gaussian quadrature.

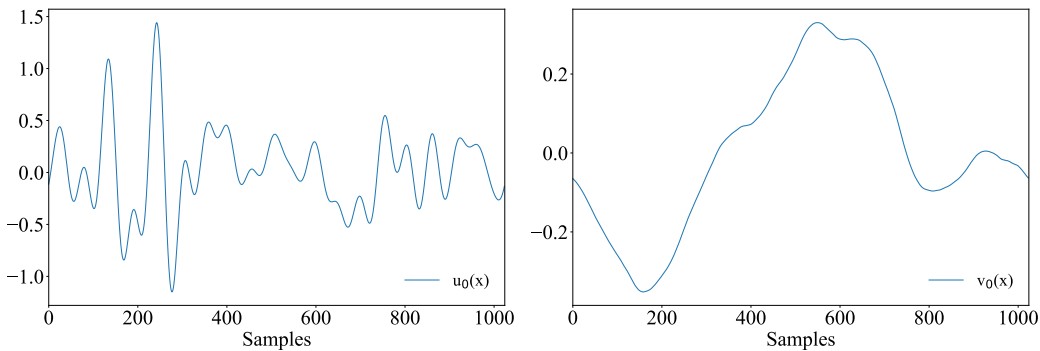

Figure 7: The sample of the initial conditions at (U-Rand, V-GRF).

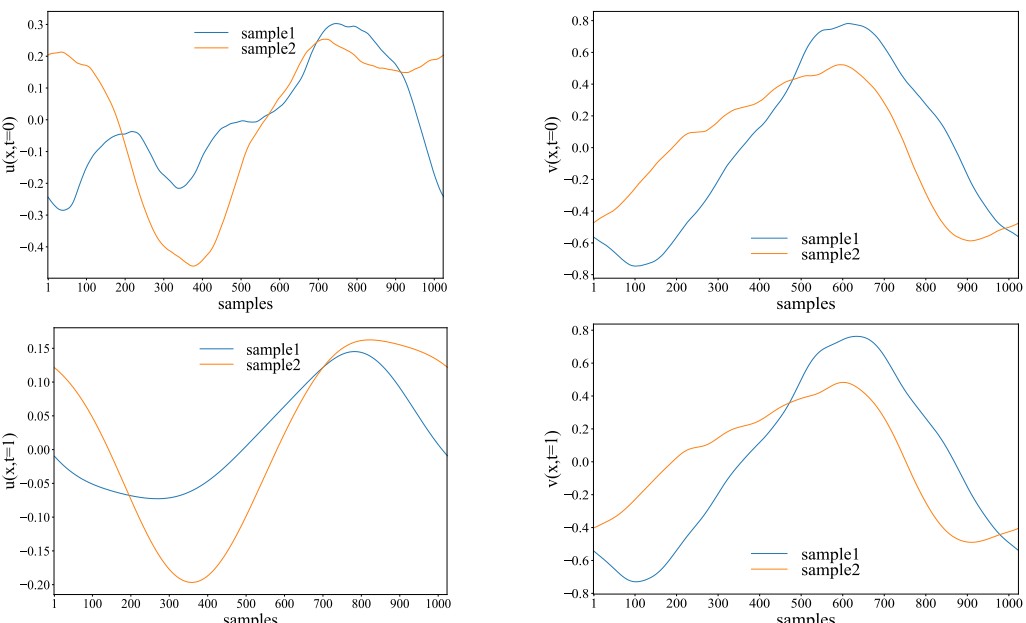

Figure 8: The input/output samples at the initial conditions (U-GRF, V-GRF).

# E    INITIAL STATES SAMPLES

In this section, we use Fig. 7 to illustrate the initial states (u-rand and v-grf) for Gray-scott equations; use Fig. 8 to exhibit the different initial states (u-grf and v-grf) and solutions for Gray-scott equations; use Fig. 9 to show the samples of $\rho(x, t)$ and $\phi(x, t)$ at different time $t$ in our non-local MFG case.

# F    ADDITIONAL RESULTS

## F.1    ADDITIONAL RESULTS FOR GRAY-SCOTT (GS) EQUATIONS

In this section, we provide more results on the $L2$ errors with different initial conditions on Table 3.

## F.2    BELOUSOV-ZHABOTINSKY (BZ) EQUATIONS

Adapted from the Belousov–Zhabotinsky dynamic system, the coupled Belousov-Zhabotinsky (BZ) equations in (21) describe a reaction-diffusion process with three species Driscoll et al. (2014). For a

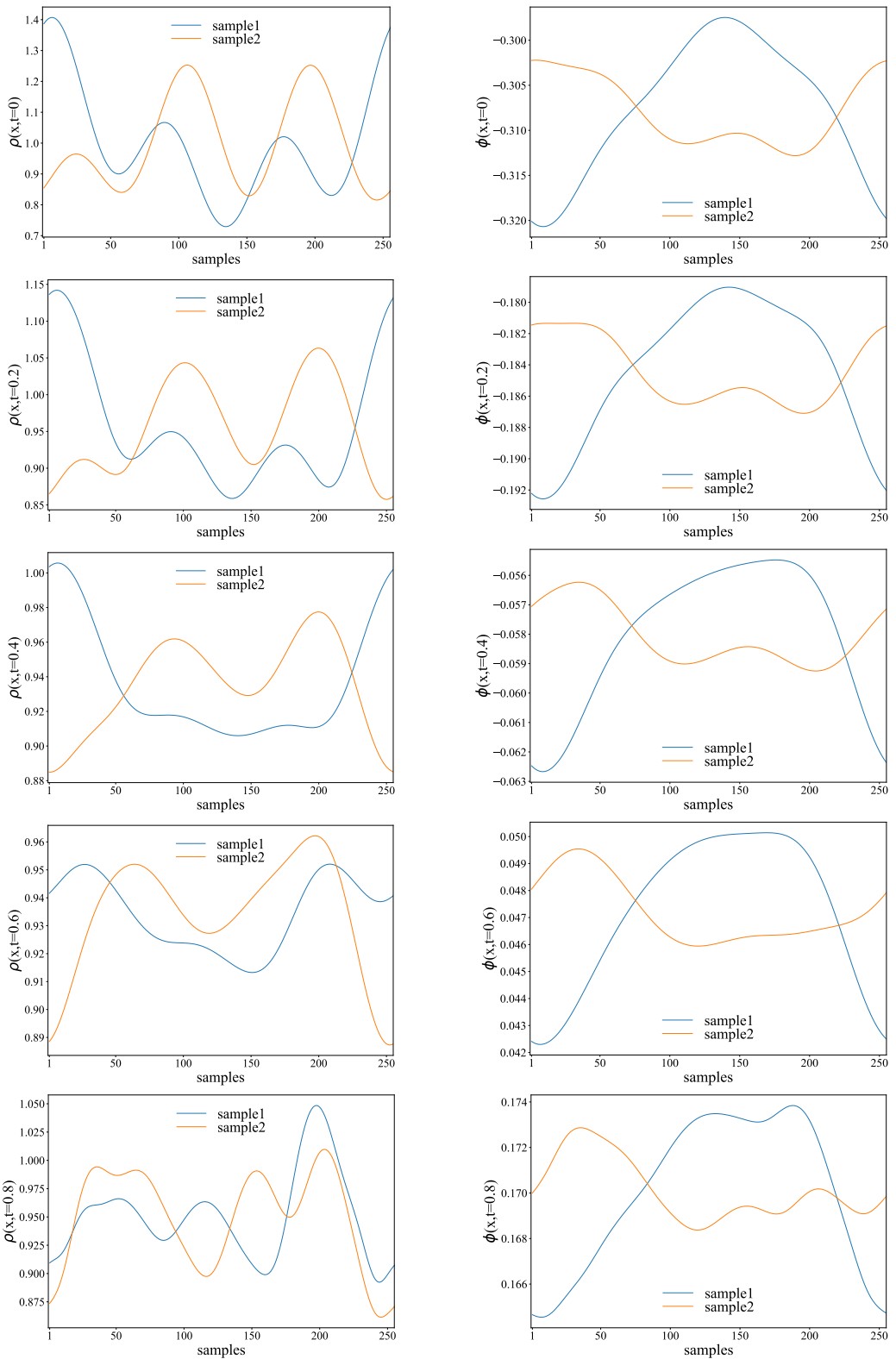

Figure 9: The sample input/output for $\rho$ and $\phi$. The top two figures are the initial conditions of $\rho$ and $\phi$ as the input. The other figures are the output of $\rho$ and $\tau$ with different time $t$.

| Models | s=256 | | s=512 | | s=1024 | |
|--------|-------|-------|-------|-------|--------|-------|
| | u | v | u | v | u | v |
| CMWNO | **0.00937** | **0.00856** | **0.00866** | **0.00726** | **0.00819** | **0.00769** |
| MWT$_s$ | 0.09353 | 0.05203 | 0.09137 | 0.05121 | 0.09381 | 0.05161 |
| MWT$_c$ | 0.02997 | 0.01804 | 0.02799 | 0.01832 | 0.02619 | 0.01866 |
| FNO$_c$ | 0.02496 | 0.01511 | 0.02433 | 0.01539 | 0.02335 | 0.01516 |
| Padé$_c$ | 0.04605 | 0.01449 | 0.04605 | 0.01479 | 0.04747 | 0.01509 |

Table 3: Gray–Scott (GS) equation benchmarks for different input resolution $s$ at initial condition (U-Rand,V-GRF). The relative $L2$ errors are shown for each model.

given field $u(x,t)$; $v(x,t)$; $w(x,t)$, the BZ coupled equations take the form:

$$\begin{aligned}
\partial_t u(x,t) &= \epsilon_1 \partial_{xx} u(x,t) + u + v - uv - u^2, & x \in (0,1), t \in (0,0.2] \\
\partial_t v(x,t) &= \epsilon_2 \partial_{xx} v(x,t) + w - v - uv, & x \in (0,1), t \in (0,0.2] \\
\partial_t w(x,t) &= \epsilon_3 \partial_{xx} w(x,t) + u - 2, & x \in (0,1), t \in (0,0.2]
\end{aligned} \tag{21}$$

where $\epsilon_1 = 5 \times 10^{-2}, \epsilon_2 = 5 \times 10^{-2}, \epsilon_3 = 2 \times 10^{-2}$. Our goal is to learn the operators mapping the initial condition of each variable to the solution with the interference of other variables. The initial conditions are generated by using the smooth random functions (Rand) in *chebfun* package (Driscoll et al., 2014). For the initial conditions $u(x,0)$; $v(x,0)$; $w(x,0)$, we set the parameters $\gamma = 0.3; 0.2; 0.1$ respectively. Given the initial conditions, we solve the equations using the fourth-order stiff time-stepping scheme named as *ETDRK4* (Cox & Matthews, 2002) with a resolution of $2^{10}$, and sub-sample this data to obtain the datasets with the a resolution of $2^8$. The results of the experiments on BZ coupled equations with different resolutions (i.e., s=256,1024) are shown in Table 4.

| Models | s=256 | | | s=1024 | | |
|--------|-------|-------|-------|--------|-------|-------|
| | u | v | w | u | v | w |
| CMWNO | **0.02306** | **0.02727** | **0.02412** | **0.01994** | **0.02505** | **0.02481** |
| CFNO | 0.04294 | 0.05738 | 0.05856 | 0.04087 | 0.06056 | 0.06148 |
| MWT$_s$ | 0.28703 | 0.34383 | 0.51297 | 0.27376 | 0.33410 | 0.53644 |
| MWT$_c$ | 0.04654 | 0.06489 | 0.07371 | 0.04723 | 0.06537 | 0.07287 |
| FNO$_c$ | 0.03575 | 0.09662 | 0.09302 | 0.03611 | 0.09371 | 0.09119 |
| Padé$_c$ | 0.04361 | 0.06766 | 0.08508 | 0.04479 | 0.06960 | 0.08486 |

Table 4: Belousov–Zhabotinsky (BZ) equation benchmarks for different input resolution s. The relative $L2$ errors are shown for each model.

To handle multiple coupled variables, we apply the dice strategy referring to Fig. 6 to mimic the interaction between $u$; $v$; $w$. As we can see in the results, our CMWNO still achieves the new state-of-the-art.

|             | $\mathrm{MWT}_s$ | $\mathrm{MWTW}_c$ | $\mathrm{Pade}_c$ | $\mathrm{FNO}_c$ | CMWNO  |
|-------------|--------|---------|---------|---------|--------|
| Parameters  | 508754 | 254377  | 143957  | 287425  | 508754 |

Table 5: Comparison of model's parameters.

## G    MODEL COMPARISON

Table 5 compares in detail the data of parameters from CMWNO and baselines. Although the number of parameters of our model is about twice as many as the second best model (FNO), the performance of our model is improved by 57.68% and 61.41%, separately. In solving the coupled PDE problem, our model is the optimal choice in terms of performance and power balance. One of our future efforts is to improve model efficiency, which we leave as the further work.

## H    DISCUSSION ON PINN

We note that neural operators and PINN are two recent deep learning approaches that have gathered the tractions in the community. The PINN combines the advantages of data-driven machine learning and physical modeling to train a model that automatically satisfies physical constraints with insufficient training data and has comparable generalization performance to predict important physical parameters of the model while ensuring accuracy. One can incorporate the differential form constraints from PDEs into the design of the loss function of the neural network with automatic differentiation techniques in deep neural networks. In addition, PINN cannot be used directly in a complete data-driven scenario without an exact PDE structure, and the PDE function is hard to be decided in the wild applications, However, one can take a compromise approach by relying on a specific PDE (such as (Connors et al., 2009)) to design its loss function and using it on different input functions, which should be undesirable.

## I    DEFAULT NOTATION

| | |
|---|---|
| $a$ | A scalar (integer or real) |
| $T_i$ | Generic operator |
| $A, B, C, \bar{T}$ | neural networks |
| $\mathbb{A}$ | A set |
| $\mathbb{R}$ | The set of real numbers |
| $\kappa_i$ | The kernel in the opeartor |
| $[a, b]$ | The real interval including $a$ and $b$ |
| $(a, b]$ | The real interval excluding $a$ but including $b$ |
| $\mathbf{V}_n^k$ | $\{f \mid f$ are polynomials of degree $< k$ defined over interval $(2^{-n}l, 2^{-n}(l+1))$ for all $l = 0, 1, \ldots, 2^n - 1$, and assumes 0 elsewhere$\}$ |
| $\mathbf{W}_n^k$ | Orthogonal space to $\mathbf{V}_n^k$ such that $\mathbf{W}_n^k \bigoplus \mathbf{V}_n^k = \mathbf{V}_{n+1}^k$ |
| $T_{*,*l}^*$ | Coefficients according to operator |
| $U_{*,*l}^*, V_{*,*l}^*$ | Coefficients according to input $u^*$ and $v^*$ |
| $P_n$ | Projection operator such that $P_n : \mathcal{H}^{s,2} \to \mathbf{V}_n^k$ |
| $Q_n$ | Projection operator such that $Q_n : \mathcal{H}^{s,2} \to \mathbf{W}_n^k$ |
| $L$ | Coarsest scale of the multiwavelet transform |

