# OpenReview forum: "Coupled Multiwavelet Operator Learning for Coupled Differential Equations"
_ICLR.cc/2023/Conference — ICLR 2023 poster_

### Official Review · Reviewer_k8FT · 2022-10-21

**Confidence:** 2
**Correctness:** 4
**Technical Novelty And Significance:** 3
**Empirical Novelty And Significance:** 3
**Recommendation:** 6

**Clarity, Quality, Novelty And Reproducibility:**

The manuscript is well-written and the experiments are reproducible. I believe that the neural operator architecture as well as the dice strategy for training neural operators are two novel contributions.

**Strength And Weaknesses:**

I am not aware of any existing neural operator approaches for coupled PDEs, and the use of the multiwavelet transform is very reasonable in this context. The dice strategy is a nice observation to deal with the imbalance update problem that could occur for each kernel.  The authors compare their approach with many other reasonable neural operators, and some improved accuracy is observed.

There's an assumption that one can write down decoupled solution operators that solve for each component of the coupled PDEs in (5). I find it hard to assess if this is a mild assumption or not.

**Summary Of The Paper:**

The manuscript presents a neural operator approach for solving coupled PDEs, motivated by multiwavelets. The authors demonstrate some improved accuracy on Gray-Scott and nonlocal mean field game problems. The multiwavelet transform is used to represent the coupled information from the coupled PDEs and a dice strategy is employed to mimic the interaction of the information during the training process.

**Summary Of The Review:**

The authors are proposing a strategy for solving coupled PDEs that sounds reasonable, and uses a multiwavelet discretization of the spatial domain. There are some improvements in accuracy for the Gray-Scott and nonlocal mean field game problems. Given the assumption in  (5), I find it difficult to assess how well this approach will work as a general approach for coupled PDEs.

---

> ### Author Response · Authors · 2022-11-12
> **Response to Reviewer k8FT**
>
> Thank you for your time and helpful comments to make our paper more precise! We now take the opportunity to answer your concern on the assumption:
>
> Decoupling requires manual scientific effort by understanding the coupling behavior which usually varies from different coupled PDEs. Thus, it should be meaningful and cost-effective if a general approach can be proposed to solve coupled PDE by pure data-driven methods. In this paper, we introduce a well-designed coupled multiwavelets neural operator to decouple and then solve coupled PDEs automatically without the knowledge of the equations. The intuition behind this work is that one can decouple the coupled kernels into two parts: one part is the group of independent operators which only contains the separate information, and the other part is the interaction part containing the “coupled” information between the kernels (equations 4 and 5). Given specific decoupled solution equations are non-trivial and hard to guarantee correctness. However, approximating such decoupled equations is possible with the help of machine learning methods. Then the question becomes how we can efficiently represent the kernels and model the interactions between the kernels. Motivated by the representation capability of multiwavelets space, we propose to use the first-order coefficients in the multiwavelet space to model the coupled information between the kernels.
>
>  What’s more, we demonstrate the generalization capabilities of CMWNO on solving coupled PDEs empirically by establishing new state-of-the-art on three different types of coupled PDEs (i.e., Gray-Scott (GS) equations, the non-local mean field game (MFG) equations, and Belousov-Zhabotinsky (BZ) equations with three coupled variables). The updated results can be found in our revised version in Table 1, 2 and 6.

---

### Official Review · Reviewer_Zj6R · 2022-10-25

**Confidence:** 4
**Clarity, Quality, Novelty And Reproducibility:** 1. Clarity

    The presentation is c…
**Correctness:** 3
**Technical Novelty And Significance:** 2
**Empirical Novelty And Significance:** 2
**Recommendation:** 6

**Strength And Weaknesses:**

Strength

1. Main objective (to solve coupled PDEs in the regression setting) is clearly formulated.
2. Most of the main claims are reasonably substantiated (for exceptions see below).
3. Most ideas are clearly explained (for exceptions see below).

Weaknesses

1. Architecture is not described in detail.

    In Section 3 (page 6) authors describe neural networks A, B, C, and T used in multiwavelet neural operators as follows
    ``In our proposed model, for each operator, the neural networks A, B, and C use single-layered convolutional neural networks while T uses a single linear layer.``

    This description is incomplete. What are the sizes of convolutions, the strides the number of input and output channels (features)? Are there any nonlinear activations and what are those? Are there multiple multiwavelet kernels (cells) involved and how many of them?

2. The proposed model is a minor modification of [1].

    From the description in the main text and the parameter count in Table 6 (supplementary materials), it looks like the proposed architecture is exactly the same as in [1]. The contribution then is a way how the known architecture can be applied to coupled equations.

3. Awkward notation

    In many places, authors use backslash / to refer the reader to distinct cases. This does not look appropriate when mathematical expressions are involved since / is also used to denote division, quotient group, and other mathematical operations.  For a concrete example see page 5, text right before equation 9.

4. The first claimed contribution is unjustified.

    ``For coupled differential equations, we propose a coupled neural operator learning scheme, named CMWNO. To the best of our knowledge, CMWNO is the first work using pure data-driven method to solve coupled differential equations.``

    It is not clear what the authors mean. Any work that solves the Navier-Stokes equation using machine learning deals with coupled PDEs since velocities are coupled. See [2] for a concrete example. Besides that, there are works on multiphase flows [3], weather forecasts [4], e.t.c. Essentially, there are hundreds if not thousands of papers dealing with coupled PDEs in regression and physics-informed setups.

5. The choice of a physics-informed neural network (PINN) as a benchmark is questionable.

    PINN provides a way to solve PDE given initial data. As the present reviewer understands vanilla PINN takes only coordinates for input and produces the solution. That is, PINN is not a neural operator in contrast with all other considered architectures. How PINN is applied precisely in this context? Is it retrained for each new input function? In the opinion of the present reviewer, PINN adds little to the understanding of how the accuracy of the presented model compares to the state-of-the-art.

[1] - Gupta G, Xiao X, Bogdan P. Multiwavelet-based operator learning for differential equations. Advances in Neural Information Processing Systems. 2021 Dec 6;34:24048-62.

[2] - Stachenfeld K, Fielding DB, Kochkov D, Cranmer M, Pfaff T, Godwin J, Cui C, Ho S, Battaglia P, Sanchez-Gonzalez A. Learned Coarse Models for Efficient Turbulence Simulation. arXiv preprint arXiv:2112.15275. 2021 Dec 31.

[3] - Wen G, Li Z, Azizzadenesheli K, Anandkumar A, Benson SM. U-FNO—An enhanced Fourier neural operator-based deep-learning model for multiphase flow. Advances in Water Resources. 2022 May 1;163:104180.

[4] - Pathak J, Subramanian S, Harrington P, Raja S, Chattopadhyay A, Mardani M, Kurth T, Hall D, Li Z, Azizzadenesheli K, Hassanzadeh P. Fourcastnet: A global data-driven high-resolution weather model using adaptive Fourier neural operators. arXiv preprint arXiv:2202.11214. 2022 Feb 22.

**Summary Of The Paper:**

The authors proposed a modification of multiwavelet architecture introduced in [1] for the case of coupled PDEs. Comparisons with the Fourier Neural Operator [2], two versions of the original multiwavelet operator [1], the Pade exponential model [3], and Physics-informed neural networks [4] show that the proposed modification improves relative L2 error.


[1] - Gupta G, Xiao X, Bogdan P. Multiwavelet-based operator learning for differential equations. Advances in Neural Information Processing Systems. 2021 Dec 6;34:24048-62.

[2] - Li Z, Kovachki N, Azizzadenesheli K, Liu B, Bhattacharya K, Stuart A, Anandkumar A. Fourier neural operator for parametric partial differential equations. arXiv preprint arXiv:2010.08895. 2020 Oct 18.

[3] - Gupta G, Xiao X, Balan R, Bogdan P. Non-linear operator approximations for initial value problems. In: International Conference on Learning Representations 2021 Sep 29.

[4] - Raissi M, Perdikaris P, Karniadakis GE. Physics-informed neural networks: A deep learning framework for solving forward and inverse problems involving nonlinear partial differential equations. Journal of Computational Physics. 2019 Feb 1;378:686-707.

**Summary Of The Review:**

Overall, the paper is well-written and organized and mostly well-explained. Provided experiments are of good quality and models chosen for benchmarks include state-of-the-art architectures. The approach given by the authors is empirically justified and illustrated with (vague) theoretical considerations.

The main concern of the present reviewer is that the paper does not contain significantly novel techniques. It only proposes a minor correction of architecture from [1].


[1] - Gupta G, Xiao X, Bogdan P. Multiwavelet-based operator learning for differential equations. Advances in Neural Information Processing Systems. 2021 Dec 6;34:24048-62.

---

> ### Author Response · Authors · 2022-11-12
> **Response to Reviewer Zj6R**
>
> Thank you for the constructive comments and insightful suggestions for improving our paper’s presentation! The point-wise response to your concerns is as follows:
>
>
> 1. Thank you for the clear questions about the detailed description of the architecture. We have added the detailed architecture in supplementary B (Page 14-15). Specifically, for two coupled variables $u$ and $v$, the trainable neural networks layer in the decomposition stage is composed of $\{A_u, B_u, C_u,\bar{T}_u\}$ and $\{A_v, B_v, C_v,\bar{T}_v\}$. We use two cascading neural network layers with normal Xavier initialization to handle the coupled PDEs in our experiments.  Moreover, $\{A_u, B_u, C_u\}$ are all one-layer CNNs with the ReLU activation function followed by a linear layer, where CNN's kernel size equals 3, stride equals 1, and padding size equals 1. The input channel of CNN equals the feature number and the output channel is set to 128 in all the experiments. In addition, $\{\bar{T}_v\}$ is a single $k\times k$ linear layer with $k = 4$ suggested by [1].
>
>      In addition, we provide a detailed general version of our dice strategy in Appendix (Fig.6). For $n$ coupled variables, we have $n$ multiwavelet kernels that need to be updated and $m$ paths that can be selected during the training process, where $m=\mathcal{A}_n^n$ and $\mathcal{A}$ is the function of all permutations.
>
> [1] Gupta, G., Xiao, X., & Bogdan, P. (2021). Multiwavelet-based operator learning for differential equations. Advances in Neural Information Processing Systems, 34, 24048-24062.
>
> 2. Thank you for raising your concerns about the architecture modification. We want to clarify that the main novelty of this work is combining the multiwavelets and coupled PDEs by designing separate kernels and proposing the dice strategy to deal with the coefficients in multiwavelets space, especially during the decomposition and reconstruction processes. The problem formulation of solving coupled PDEs by decoupling the coupled variables in machine learning-oriented fields is also original and inspirable to the best of our knowledge. Based on our knowledge of multiwavelets and multiwavelets-based operator learning, we assume that decoupling and then solving the coupled PDEs by the complete data-driven model can be achieved by multiwavelet operator learning with the good bases, which allow us to represent the functions and kernels using the first-order coefficients of multi-scale scaling/ wavelet functions. In this work, we also found that decoupling in the multiwavelet space helps a lot for solving coupled equations, which is evident from the results in the experiments section.
> Moreover, we would like to address that the dice strategy is especially helpful to mimic the interaction of the information during the training process. In the additional experiments, we test CFNO on GS, BZ coupled PDEs, as well as non-local MFG, and CFNO beats FNO in most cases, which indicates that our proposed dice strategy is beneficial for decoupling. It is worth noting that the CMWNO outperforms CFNO in all the cases, which indicates that the multiwavelet space is more beneficial for the decoupling procedure than Fourier space.
> The partial results of GS PDEs are shown below:
>  | Models |         s=256         |         s=512         |         s=1024        |
> |--------|:---------------------:|:---------------------:|:---------------------:|
> | CMWNO  | **0.00468** (u) **0.00464**(v) | **0.00492**(u) **0.00434** (v) | **0.00471** (u) **0.00450**(v) |
> | CFNO   | 0.01371(u) 0.00654(v) | 0.01345(u) 0.00643(v) | 0.01421(u) 0.00619(v) |
>
>      We have also updated the results of CFNO in Table 2 (Page 8) and Table 6 (Page 20) in the revised version, and we will update the results of all the other experiments in the revised version soon.
>
> 3. Thank you for pointing out the confusion about the notation. To avoid confusion, we have modified the backslash to a clearer expression. For example, we modified $\{A_{u\/v} ,B_{u/v}, C_{u/v} \}$   to
>  $\{A_{\[u,v\]}, B_{\[u,v\]}, C_{\[u,v\]} \}$.

---

> > ### Author Response · Authors · 2022-11-12
> > **Cont.**
> >
> > 4. Thank you for bringing up the concern about the first claimed contribution. We have reformulated the statement to better reflect our contributions and results. Specifically, we made the following change: “To the best of our knowledge, CMWNO is the first neural operator work using a pure data-driven method to decouple and then solve coupled differential equations.” We also want to clarify that, different from the previous work focusing on solving PDEs, we focus on the term “coupled” and aim to decouple the coupled PDEs using the complete data-driven neural operator.
> >
> >     Thank you for sending us the related papers. The works take advantage of FNO’s ability to efficiently solve PDEs to design models for applications in real chaotic systems, which contain the coupled variables. However, none of these works discuss or focus on decoupling the coupled terms, which is the novel part of our work. To reach a broader audience, we have added them into the references and discussed them in the related work section (Page 14) as follows:
> >
> >      “Several works take advantage of FNO’s ability to efficiently solve PDEs to design models for applications in practical chaotic systems such as turbulence simulation [1], multiphase flow simulation  [2], and weather forecasting [3].”
> >
> > [1] - Stachenfeld K, Fielding DB, Kochkov D, Cranmer M, Pfaff T, Godwin J, Cui C, Ho S, Battaglia P, Sanchez-Gonzalez A. Learned Coarse Models for Efficient Turbulence Simulation. arXiv preprint arXiv:2112.15275. 2021 Dec 31.
> >
> > [2] - Wen G, Li Z, Azizzadenesheli K, Anandkumar A, Benson SM. U-FNO—An enhanced Fourier neural operator-based deep-learning model for multiphase flow. Advances in Water Resources. 2022 May 1;163:104180.
> >
> > [3] - Pathak J, Subramanian S, Harrington P, Raja S, Chattopadhyay A, Mardani M, Kurth T, Hall D, Li Z, Azizzadenesheli K, Hassanzadeh P. Fourcastnet: A global data-driven high-resolution weather model using adaptive Fourier neural operators. arXiv preprint arXiv:2202.11214. 2022 Feb 22.
> >
> > 5. Thank you for pointing out your concern about the choice of PINN as a benchmark. We agree with the reviewer that PINN cannot compare directly with neural operators, so to maintain the consistency of the theme of the paper, we have moved the discussion on PINN to Supplementary H (Page 21). Previously, we have already added all recent successful neural networks in the baseline and adding PINN shows that CMWNO performs better than all recent deep-learning approaches for coupled PDEs. However, the PDE function is hard to be decided in the wild applications. Therefore, we have taken a compromise approach by relying on a specific PDE to design its loss function and using it on different input functions.
> >
> > By the way, we have uploaded the necessary library to run our code, and we will public the code after being accepted.

---

### Official Review · Reviewer_iKvj · 2022-10-30

**Confidence:** 3
**Clarity, Quality, Novelty And Reproducibility:** Ok for clarity and reproducibility. G…
**Correctness:** 3
**Technical Novelty And Significance:** 3
**Empirical Novelty And Significance:** 3
**Recommendation:** 6

**Strength And Weaknesses:**

Strengths

- The high-level motivation combining multiwavelets and coupled PDEs is very interesting and original (to the best of my knowledge)
- The results are significant when compared to standard learning-based baseline methods


Weaknesses and Questions

- My main concern is the assumptions leading to equations 4 and 5. I am not fully on board with why the multiwavelet space automatically guarantees decoupling? I presume this assumption is crucial since the construction of the architecture for CMWNO is hinged on such a separation and solved in an iterative fashion (i.e. fix the coupling dependence and solve for the kernel and vice versa). Despite this, what is the need for multiwavelets at this step? and not simple wavelets or the Fourier transform?  (which can also lead to an interesting baseline)
- Even though the improvement over shown baselines is good, I am not sure if it is a super convincing comparison since most of these approaches do not seem to be designed for coupled PDE’s. I do feel that the concatenation approach to make these methods work for the coupled system is a fair attempt, but I am not too surprised they don’t work well.
- The paper focuses on coupling with only two variables. How can the proposed framework be extended for a more sophisticated coupling PDE system with many variables? especially the dice strategy?
- Broadly, it appears that the positioning of this paper appeals more to a core PDE audience (giving them a new tool with neural networks for complex PDEs like Gray Scott equations, etc) rather than a representation learning conference. However, this paper is not entirely out of place for ICLR, but I feel that the background and motivation for coupled PDEs and Multiwavelets can be made more accessible. For example, a visual schematic, or some visualizations depicting the use cases for multiwavelets and coupled PDEs would go a long way to make everything convincingly self-contained.

**Summary Of The Paper:**

This paper proposes using the multiwavelets framework to solve coupled partial differential equations in a data-driven manner. Multiwavelets are a generalization of standard scalar wavelets but instead use more than one scaling function and the multiwavelet transform is not a straightforward separable extension of the wavelet transform but contains interaction coefficients due to this multivariable choice.

At the other hand, in scenarios of coupled partial differential equations in which more than one differential equation needs to be solved with interacting terms, the greens function kernels are not independent and hence not amenable to be solved using well-known neural operators like PINNs, Fourier Neural Operators (FNO), etc in a straightforward manner.

The authors combine the two scenarios and propose the coupled multiwavelet neural operator (CMWNO). Experiments are demonstrated for 2 problems - Gray Scott Equations and the Mean Field Game Problem. The results show good improvements over baseline neural network approaches by a significant margin.


**Summary Of The Review:**

Overall, I think this paper is quite decent in its methodology and evaluation, and contains an interesting combination of multiwavelets and coupled PDEs. Despite that, I find some ambiguity in the proposed framework, especially regarding the architecture design and assumptions leading to Eq 4 and 5. I’ll wait for the rebuttal to make a more informed opinion, but as of now, it is a borderline rating from me.

---

> ### Author Response · Authors · 2022-11-12
> **Response to Reviewer iKvj**
>
> We pretty much appreciate your recognition of our work by pointing out that our idea of combining multiwavelets and coupled PDE is very interesting and original! We now take the opportunity to clarify your concerns:
>
> 1. Thank you for bringing up your concerns about (a). our assumption on equations 4 and 5 (b). why do we need multiwavelets instead of simple Wavelet or Fourier transform.
>
>     (a). Since decoupling requires manual scientific effort by understanding the coupling behavior. To develop a complete data-driven approach as well as being data efficient, we are required to put mild assumptions. In our paper, the assumption is that one can decouple the coupled PDEs by decoupling the coupled kernels of different variables into two parts: one part is the group of independent kernels which only contains the separate information, and the other part is the interaction part containing the “coupled” information, which leads to equations 4 and 5. Then the question becomes how we can efficiently represent the kernels and model the interactions between the kernels. Motivated by the representation capability of multiwavelets, we propose to use the first-order coefficients in the multiwavelet space to model the coupled information between the kernels.
>
>     We also want to clarify that we do not want to overclaim that multiwavelet space can automatically guarantee decoupling in our paper. We instead argue that the multiwavelets are beneficial for decoupling the kernels in Wavelet space because the multiwavelets provide useful bases allowing us to represent the functions and kernels using the first-order coefficients of multi-scale scaling/ wavelet functions. In this work, we found that the multiwavelet transform helps a lot, which is evident from the results in the experiments section.
>
>     (b). Here we answer the question of why we need multiwavelets instead of simple wavelets and Fourier transform. For the simple wavelets, they are also able to provide the operational matrix to solve PDEs, however, they require a large number of coefficients for functional approximation  [1,2,3,4]. In addition, the simple discrete wavelet transform is proposed to deal with functions defined over the entire real line and there are ‘boundary problems’ when applying the wavelets for finite interval functions [1,5]. By dealing with the transformation of scaling/wavelet coefficients in multiwavelet space, we have not encountered the issues. Compared with the Fourier transform, the multiwavelets sparsify the kernel and show a better performance than Fourier transform especially on high-frequency data [6,7]. In addition, we appreciate your insightful suggestion on adding an “interesting” baseline of decoupling the kernels with Fourier transform. Therefore, we also conduct experiments where we create multiple kernels represented in Fourier space and apply the dice strategy to decouple the kernels during Fourier transform (CFNO). We test CFNO on the GS equation with different resolution, the non-local MFG to compare it with CMWNO, and our CMWNO still achieve the new state-of-the-art, partial results are shown below,
>
>       | Models |         s=256         |         s=512         |         s=1024        |
>       |--------|:---------------------:|:---------------------:|:---------------------:|
>       | CMWNO  |  **0.00468** (u) **0.00464** (v) | **0.00492** (u) **0.00434** (v) | **0.00471** (u) **0.00450** (v) |
>       | CFNO   | 0.01371(u) 0.00654(v) | 0.01345(u) 0.00643(v) | 0.01421(u) 0.00619(v) |
>
>      Table 1. Relative $L2$ errors on Gray–Scott equation.
>
>
>     | Models |         t=0.2         |         t=0.4         |         t=0.6         |
>     |--------|:---------------------:|:---------------------:|:---------------------:|
>     | CMWNO  | **0.00083** ($\rho$) **0.00073** ($\phi$) | **0.00154** ($\rho$) **0.00252** ($\phi$) | **0.00543** ($\rho$) **0.00467**($\phi$) |
>     | CFNO   | 0.00767($\rho$) 0.00162($\phi$) | 0.00890($\rho$) 0.00638($\phi$) | 0.02011($\rho$) 0.01029($\phi$) |
>
>     Table2. Relative $L2$ errors for predicting $\rho(x,t)/\varphi(x,t)$
>
> The complete results of Table 1,  2 can be found in our revised version.  We will also update the results of CFNO of all the experiments in the paper as soon as possible.
>
> [1]. Heydari, M. H., Hooshmandasl, M. R., & Mohammadi, F. (2014). Legendre wavelets method for solving fractional partial differential equations with Dirichlet boundary conditions. Applied Mathematics and Computation, 234, 267-276.
>
> [2]. Chou, K. C., & Guthart, G. S. (2000). Representation of Green's function integral operators using wavelet transforms. Journal of Vibration and Control, 6(1), 19-48.
>
> [3]. Amaratunga, K., & Williams, J. R. (1993). Wavelet based Green's function approach to 2D PDEs. Engineering Computations, 10(4), 349-367.

---

> > ### Author Response · Authors · 2022-11-12
> > **Cont. 1**
> >
> > [4]. Razzaghi, M., & Yousefi, S. (2001). The Legendre wavelets operational matrix of integration. International Journal of Systems Science, 32(4), 495-502.
> >
> > [5] Hasumi, R., & Kajita, Y. (2018). Boundary problem and data leakage: A caveat for wavelet-based forecasting.
> >
> > [6] Alpert, B. K. (1993). A class of bases in L^2 for the sparse representation of integral operators. SIAM journal on Mathematical Analysis, 24(1), 246-262.
> >
> > [7] Gupta, G., Xiao, X., & Bogdan, P. (2021). Multiwavelet-based operator learning for differential equations. Advances in Neural Information Processing Systems, 34, 24048-24062.
> >
> >
> > 2. Thank you for your comments on the baselines. The existing neural operator learning approaches are not exactly designed for decoupling the coupled PDEs, and there is no general architecture for neural operators to decouple the coupled PDEs to the best of our knowledge. Therefore, we use the concatenation approach to make an alternative and relatively fair comparison with exist approches. We also appreciate and follow your suggestion of decoupling the kernels in the Fourier space during the Fourier transform, and we have discussed the experiments above. Decoupling the kernels in multiwavelets space can still achieve the new state-of-the-art in all our experimental settings.
> >
> > 3. Thank you for raising your concerns about the scalability of our model for more sophisticated coupling PDEs. We have mentioned in the paper that our model is scalable corresponding to the number of coupled variables in the Experiments section on Page 6. To make it clearer, we have added a more general illustration for the dice strategy in Supplementary Fig.6 (Page 14). Specifically, for each sample of coupled variables, one only needs to go through a specific path (round diagonal corner rectangle in Fig.6) at one training time. For example, we get dice equal to $1$ in the case of Fig.6, so that this specific sample will go through Path 1. Inside each path, the order of updating is from left to right, where the darker block indicates the operator we want to update and the lighter blocks provide decomposition information from the fixed operator. Note that the updated operator can help to update other operators at the same stage that occurs after the specific operator. Thus, for $n$ variables, we have $n$ operators that need to be updated and $m$ paths that can be selected, where $m=\mathcal{A}_n^n$ and $\mathcal{A}$ is the function of all permutations.
> >
> >
> >     In addition, we also conducted experiments on Belousov–Zhabotinsky coupled equations with 3 coupled variables to extend the application scenario. The detailed experiments can be found in Supplementary F.2 (Page 20), where CMWNO can beat all the competitive baselines. Here we put the results for resolution s = 256. The results of higher resolution (i.e., s=1024) can be found in Supplementary F.2 (Page 20).
> >
> >     | Models |    u    |    v    |    w    |
> >     |--------|:-------:|:-------:|:-------:|
> >     | CMWNO  |  **0.02306**  |   **0.02727** |   **0.02412** |
> >     | CFNO   | 0.04294 | 0.05738 | 0.05856 |
> >     | MWTs   | 0.28703 | 0.34383 | 0.51297 |
> >     | MWTc   | 0.04654 | 0.06489 | 0.07371 |
> >     | FNOc   | 0.03575 | 0.09662 | 0.09302 |
> >     | Padec  | 0.04361 | 0.06766 | 0.08508 |
> >
> >     Table3.  Relative $L2$ error on Belousov–Zhabotinsky (BZ) equation.

---

> > > ### Author Response · Authors · 2022-11-12
> > > **Cont. 2**
> > >
> > > 4. Thank you for telling us your concerns about the positioning of our paper. In the past several years, we have been encouraged to see that increasing neural operator-related works provide promising results in multiple downstream machine learning tasks and are accepted by top ML conferences such as FNO (ICLR), Pade (ICLR), MWT (NeurIPS), etc.  Specifically, this work aims to learn the representation of initial data that can help to solve the coupled PDE via a well-designed coupled neural operator structure.  Guided by [1] (12099 citations by Nov.8), in order to generate a good representation, we need to build different neural operators as coupled PDE contains at least two variables that share the coupled information across equations ( **Shared factors across tasks**  section in [1]), and we also need to simplify the factor dependencies in the multiwavelet space ( **Mainfolds** and  **Simplicity of Factor Dependencies** sections in [1]). Thus, ICLR could be a suitable venue for this research line of neural operators. In addition, we would like to address that this paper aims to provide some inspiration to 2 types of audiences. Moreover, for the PDE audience, based on our knowledge, this is the first work formulating the problem of decoupling the coupled PDEs into a machine learning-orientated field; for the neural operator audience, we empirically exhibited that the neural operator has a strong ability on decoupling and solving coupled PDEs problems, which broadens the application scenario of neural operator.
> > >
> > > ​​[1] Bengio, Yoshua, Aaron Courville, and Pascal Vincent. "Representation learning: A review and new perspectives." IEEE transactions on pattern analysis and machine intelligence 35.8 (2013): 1798-1828

---

> > > > ### Author Response · Authors · 2022-11-16
> > > > **Follow Up**
> > > >
> > > > We have uploaded the revised version with the updated results as we mentioned in the response. We would appreciate it if you could let us know whether we addressed your concerns. We are also happy to discuss any further valuable problems with you.

---

> > ### Comment · Reviewer_iKvj · 2022-11-17
> > **Plain Wavelets**
> >
> > Thank you for your detailed rebuttal. I still have one lingering question - what was exactly wrong with plain wavelets? You mentioned - *For the simple wavelets, they are also able to provide the operational matrix to solve PDEs, however, they require a large number of coefficients for functional approximation [1,2,3,4]. In addition, the simple discrete wavelet transform is proposed to deal with functions defined over the entire real line and there are ‘boundary problems’ when applying the wavelets for finite interval functions [1,5]*.  Conceptually, why shouldn't this also be a problem for Fourier transform and possibly multiwavelets as well?  I acknowledge that you dont encounter a problem in practice and its great that you report it as such, but I am trying to decode a conceptual reason behind this behaviour. I am asking this to build a final opinion - especially to address the point made by reviewer Zj6R, about novelty with respect to [1]

---

> > > ### Author Response · Authors · 2022-11-18
> > > **Response to "Plain Wavelets"**
> > >
> > > Thank you for your additional questions. As we discussed in the response, a plain wavelet basis is defined on the entire line $R$, and by tensor product, on the entire $R^n$. On **compact intervals**, wavelet coefficients will suffer from shift issues when they are near the endpoint in the transformation window, known as “boundary problems” [1]. Thus, the construction of orthonormal basis vectors has to be adjusted. There are general adjustment methods in literature that work for orthonormal wavelet bases (see [2]). Such methods "fold" the definition of the orthonormal basis (ONB) vectors when approaching the boundary, but such "folding" comes with a high computational cost. Instead, multiwavelets use orthogonal polynomials which are defined over a finite interval, for example, Legendre polynomials defined over [-1,1]. Thus, multiwavelet space is **a finite interval polynomial space**. In our model, we utilize the orthonormal basis whose vectors have their support either completely included in the interval or disjoint to the interval. The multiwavelet ONB described in Supplementary C and D (pages 15, 16) is constructed to satisfy precisely this condition. The fact that we use Legendre polynomials in the construction of Multiresolution Analysis guarantees that wavelets $\Psi_j$ have $k$ vanishing moments (see equation (17)), which in turn, provide efficient representations of functions.
> > >
> > > Fourier transform based models also suffer from non-periodic functions. To deal with non-periodicity, usually zero-padding is done. Note that zero-padding is only a work around to deal with non-periodicity of a finite interval function while multiwavelets, being a finite interval projection, do not require such additional treatment.
> > >
> > > Additionally, Fourier coefficients based decoupling (in equation (4)) will result in a convolution-like structure in the function space. Therefore, we argue that an approach based on Fourier transform is not appropriate for dealing with such a **non-linear decoupling**, as we have in the coupled Gray-Scott PDEs in equation (11) (i.e., the $uv^2$ term). However, multiwavelets based decoupling allows a finer granularity through coefficients addition at individual scales (equation (10)). The flexibility offered by multi-scale decoupling provides a better way to learn the non-linear coupling as we demonstrate through empirical experiments in section 3 (CMWNO vs CFNO).
> > >
> > > [1] Hasumi, R., & Kajita, Y. (2018). Boundary problem and data leakage: A caveat for wavelet-based forecasting.
> > >
> > > [2] Cohen, A., Daubechies, I., & Vial, P. (1993). Wavelets on the interval and fast wavelet transforms. Applied and computational harmonic analysis.

---

### Author Response · Authors · 2022-11-12
**Response to all reviewers**

We sincerely thank all the reviewers for their careful reading and thoughtful suggestions/ comments on our paper. We find it encouraging that reviewers recognize our novelty in combining the multiwavelets and coupled PDEs! The reviewers’ suggestions also led us to further improve the clarity of our paper. Here we briefly summarize the proposed changes (marked in blue in our revised version) and we will address the individual reviewer’s comments accordingly.

1. **Add a new baseline.**  By creating multiple kernels learned in Fourier space and applying the dice strategy during Fourier transform, we have built the new baseline and marked it as CFNO.

2. **Additional experiments.**  We conduct experiments on Belousov-Zhabotinsky (BZ) equations with three coupled variables and the results are shown in Supplementary F.2 (Page 20).


3. **Architecture description.** We have provided a detailed illustration of our scalable dice strategy in Figure 6 (page 14). We have also added the detailed architecture description of CMWNO in Supplementary B (page 14).

4. **Additional references.** We add several additional references to the revised version such that our work can reach a broader audience.

---

### Decision · Program_Chairs · 2023-01-20

**Decision:**

Accept: poster

**Justification For Why Not Higher Score:**

The paper is a solid work, but I don't think it brings some significant novelty to the table.

**Justification For Why Not Lower Score:**

All reviewers agree in evaluation, so would I based on my own reading.

**Metareview: Summary, Strengths And Weaknesses:**

The paper proposes a way to solve coupled PDEs using multiwavelet bases and also introduces a "dice strategy".
The authors make their best to explain difficult concepts and ideas, as well as the intuition behind their approach.

Strengths: Good presentation, impressive results on several benchmarks.
Weaknesses: The overall architecture is not very straightforward and simple, and almost leans towards a specific numerical solver (i.e. one can directly solve the problem in the multiwavelet space, rather then learning something). Reimplementing the ideas and using by others could be really difficult.

**Note From Pc:**

if the above contains the word "oral" or "spotlight" please see: "oral" presentation means -> notable-top-5% and "spotlight" means -> notable-top-25%. As stated in our emails, we are disassociating presentation type from AC recommendations